# Heteroatom doping enables hydrogen spillover via H$^+$/e$^-$ diffusion pathways on a non-reducible metal oxide

Kazuki Shun [1], Kohsuke Mori [1,2] ✉, Takumi Kidawara[1], Satoshi Ichikawa [3] & Hiromi Yamashita [1,2]

Hydrogen spillover, the simultaneous diffusion of protons (H$^+$) and electrons (e$^-$) is considered to be applicable to ubiquitous technologies related to hydrogen but limited to over reducible metal oxides. The present work demonstrates that a non-reducible MgO with heteroatom Al dopants (Al–MgO) allows hydrogen spillover in the same way as reducible metal oxides. Furthermore, a H$^+$ storage capacity of this material owing to hydrogen spillover is more than three times greater than those of various standard metal oxides based on H$^+$ transport channels within its bulk region. Atomic hydrogen diffuses over the non-reducible Al–MgO produces active H$^+$-e$^-$ pairs, as also occurs on reducible metal oxides, to enhance the catalytic performance of Ni during CO$_2$ hydrogenation. The H$^+$ and e$^-$ diffusion pathways generated by the heteroatom Al doping are disentangled based on systematic characterizations and calculations. This work provides a new strategy for designing functional materials intended to hydrogen spillover for diverse applications in a future hydrogen-based society.

Hydrogen is a promising energy vector capable of storing renewable energy provides superior utilization efficiency and lower gravimetric density compared with conventional energy carriers[1–4]. Despite this, various fundamental technologies intended to hydrogen handling are not yet sufficiently mature, and thus the required infrastructure is presently unavailable[5–7]. Hydrogen spillover continues to attract significant attention due to its unprecedented functions for more than 50 years[8–10]. Its occurrence involves the cleavage of H$_2$ and following simultaneous diffusion of protons (H$^+$) and electrons (e$^-$) on a solid substrate[11,12]. This process can be regarded as the solidification of gaseous H$_2$ in conjunction with transportation of active hydrogen species. Hence, hydrogen spillover could offer a means of centralizing hydrogen storage, transportation, and utilization together with improved performance[13–15]. Reducible metal oxides, such as TiO$_2$, CeO$_2$, and WO$_3$, are recognized as promising platforms for hydrogen spillover because they contain readily reducible cations capable of

accepting e$^-$[16–18]. However, these compounds contain rare elements each with an abundance of less than 1% within Earth's crust[19]. Economically, metal oxides containing Earth-abundant elements represent more suitable platforms for hydrogen spillover. In this regard, employing non-reducible metal oxides, such as MgO and Al$_2$O$_3$, as a platform is one of candidates. Unfortunately, they inhibit the e$^-$ diffusion due to their low reducibility, and the dominant phenomenon is consequently only H$^+$ diffusion, which significantly limits the hydrogen transportation and utilization[20–22]. The design of materials based on such common elements that also allow hydrogen spillover will enable the development of innovative and sustainable technologies for a next-generation hydrogen society.

The present work demonstrates that a non-reducible Al-doped MgO (Al–MgO), which contains only Earth-abundant elements, allows hydrogen spillover in the same way as reducible metal oxides. The hydrogen spillover property of Al–MgO was evaluated in terms of

[1]Division of Materials and Manufacturing Science, Graduate School of Engineering, Osaka University, 2-1 Yamada-oka, Suita, Osaka, Japan. [2]Innovative Catalysis Science Division, Institute for Open and Transdisciplinary Research Initiatives (ICS-OTRI), Osaka University, Suita, Osaka, Japan. [3]Research Center for Ultra-High Voltage Electron Microscopy, Osaka University, Ibaraki, Japan. ✉e-mail: mori@mat.eng.osaka-u.ac.jp

H+ diffusion and e− diffusion by utilizing a variety of characterizations and theoretical calculations. In this material, four-coordinated aluminum ($Al_{Td}$) and cation vacancies ($V_{Cat}$) were evolved to provide specific H+ transport channels and allowed more than three greater H+ storage capacities than MgO and typical metal oxides. Surprisingly, the donor levels of Al allowed the concurrent diffusion of H+ and e− over Al−MgO, as in hydrogen spillover on reducible metal oxides. The spilled hydrogen on Al−MgO promoted the redox of deposited $NiO_x$ during catalytic $CO_2$ hydrogenation to increase hydrogen utilization efficiency by a factor of 11.4. This work provides a new strategy to design functional materials made of Earth-abundant elements for diverse applications in a future hydrogen-based society.

## Results

### Structural characteristics of Al−MgO

Al-doped MgO specimens having various Mg/Al ratios were synthesized by a co-precipitation procedure, and the crystal structures were analyzed by X-ray diffraction (XRD) (Fig. 1a). As the Al proportion was increased, the diffraction peak corresponding to MgO {200} shifted to higher angles as a result of a decrease in the lattice constant, related to the substitution of Mg by Al in the MgO lattice (since the ionic radii of $Mg^{2+}$ and $Al^{3+}$ are 0.720 and 0.535 Å, respectively)[23]. The specimens having Mg/Al ratios of one and two exhibited $MgAl_2O_4$ {440} diffraction peaks, resulting in phase transformation of MgO. It appears that the MgO was able to maintain its original structure up to a Mg/Al ratio of 5 without any significant phase transformation even within the nanometric area (Supplementary Fig. 1). For these reasons, the Al-doped MgO having a Mg/Al ratio of five, referred to herein simply as Al−MgO hereafter. Surprisingly, only MgO {111} peak at

36.7° was shifted to lower angles as the proportion of Al was increased, which suggests the presence of Al not only in substitutional octahedral sites ($Al_{Oh}$) but also in interstitial tetrahedral sites ($Al_{Td}$) within the Al−MgO[24,25]. It has been reported that $V_{Cat}$ produced in periclase MgO to maintain the charge balance by the substitution of trivalent $Al^{3+}$[26,27]. Considering the three times larger amount of internal strain within Al−MgO than MgO as revealed by the Williamson-hall plots, it is likely that the specific sites such as $Al_{Td}$ and $V_{Cat}$ were introduced within Al−MgO (Supplementary Fig. 2). We quantified the concentration of $Al_{Td}$ and $V_{Cat}$ within Al−MgO by detailed structural characterizations. The composition of $Al_{Oh}$ and $Al_{Td}$ were proven to be attributed at 80.8% and 19.2% within Al−MgO, respectively, with solid-state 27Al magic angle spinning nuclear magnetic resonance (27Al MAS-NMR) spectroscopy (Fig. 1b). From the inductively coupled plasma atomic emission spectroscopy (ICP-AES) measurement, the Mg:Al:O ratio of the Al−MgO was proven to be 38.2:10.0:51.8 as shown in Fig. 1c even though the stoichiometric Mg:O ratio in pristine MgO is 1:1, which indicates the as-synthesized Al−MgO is in the cation deficient state. The energy dispersive X-ray spectroscopy (EDX) analysis provided the same tendency for atomic composition as ICP-AES and indicated that the Al was distributed in the nanometric region of Al−MgO (Supplementary Fig. 3). From the results of ICP-AES, the proportion of $V_{Cat}$ at octahedral cation sites was calculated to be 9.2% within Al−MgO (Supplementary Note 1). Hence, the incorporation of Al heteroatoms generates considerable number of two specific sites such as $Al_{Td}$ and $V_{Cat}$ within Al−MgO (Fig. 2). Based on this structure, we investigated the hydrogen spillover, the coupled H+ and e− diffusion, on the Al−MgO based on the diffusion of H+ and e−.

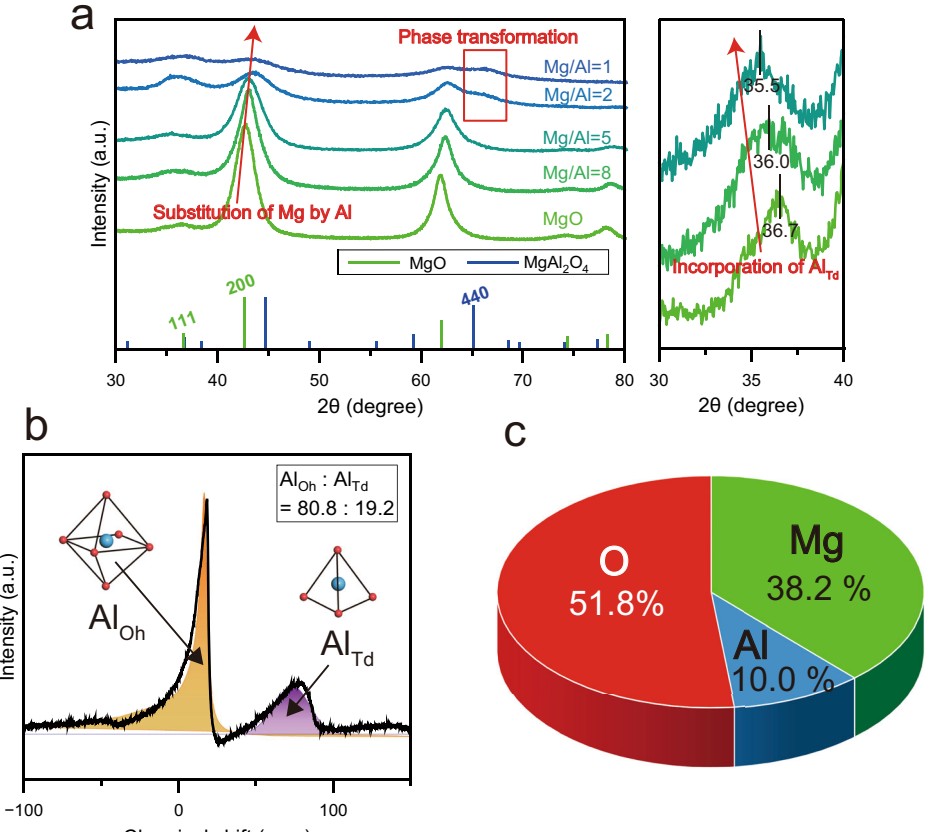

**Fig. 1 | Structural characterizations. a** XRD patterns obtained from Al-doped MgO samples having different Mg/Al ratios. Outset: an enlarged view of the MgO (111) region. **b** The 27Al solid-state MAS-NMR spectrum acquired from a Ru/Al−MgO specimen. The orange and purple shading indicate fitted peaks corresponding to $Al_{Oh}$ and $Al_{Td}$, respectively. **c** An atomic composition of the Al−MgO obtained from an ICP-AES measurement.

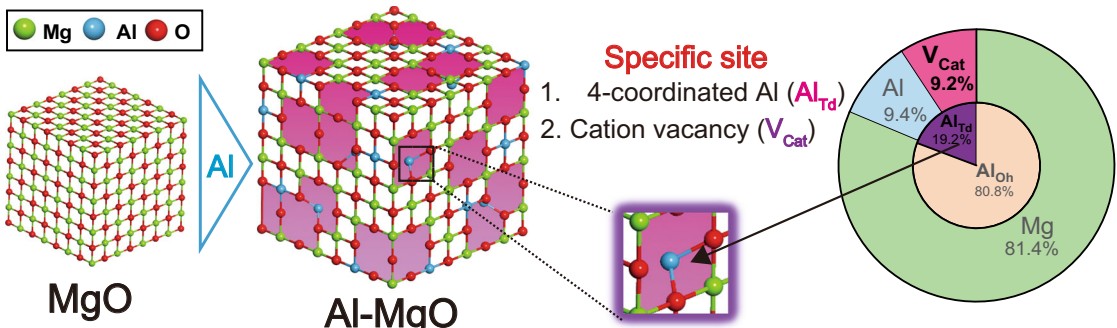

**Fig. 2 | Proposed structure of Al–MgO.** Diagram showing structural evolution of MgO following addition of Al heteroatoms as viewed from [111] direction. The Al–MgO has two types of specific sites, such as $Al_{Td}$ and $V_{Cat}$. The outer and inner regions of the circular chart provide the proportions of elements or vacancies occupying octahedral cation sites and the proportions of Al located in octahedral sites ($Al_{Oh}$) and tetrahedral sites ($Al_{Td}$) in the Al–MgO.

## H⁺ diffusion property over Al–MgO

The hydrogen spillover ability of Al–MgO was first assessed in terms of H⁺ diffusion. Ru nanoparticles were employed as dissociation sites of $H_2$. The mean diameters of Ru nanoparticles precipitated on MgO and Al–MgO were calculated to be 1.82 nm and 3.55 nm (Supplementary Fig. 4). We have previously reported that the dissociation of $H_2$ on Ru nanoparticles was a barrierless step compared with the subsequent migration steps[17], and therefore excluded the effect on the diameter of Ru nanoparticles herein. Variation in mass of Ru/Al–MgO was evaluated by thermogravimetric (TG) analyses while switching between $H_2$ and $D_2$ atmospheres (Fig. 3a). Ru/Al–MgO rapidly showed increase in its mass immediately after switching to a $D_2$ flow whereas the Ru/MgO exhibited a moderate increase. The change in mass of each sample between exposure to $H_2$ and $D_2$ could be used to calculate the extent of H⁺ storage. The proportional H⁺ storage on the Al–MgO was calculated to be 0.29 wt% and so was 3.1 times larger than that obtained using the MgO in spite of its less than half BET surface area ($S_{BET}$) (Fig. 3a). Recently, several reports have indicated that hydrogen spillover occurred in the bulk region of metal oxide platforms[28–30] and it also can endow the catalysis[31]. Considering that Al–MgO exhibited larger H⁺ storage capacity even though its $S_{BET}$ was smaller than MgO, it can be assumed that the specific H⁺ transport channels were produced within the bulk of MgO by the addition of Al. More notably, the Al–MgO showed outstanding H⁺ storage capacity compared with conventional reducible metal oxides, promising hydrogen spillover platforms. Hence, the present material would be able to store an unprecedented amount of atomic hydrogen via H⁺ diffusion even though it comprises Earth-abundant elements. Moreover, the promotional effect of Al doping on H⁺ storage capacity was confirmed for the Al-doped MgO with various Mg/Al compositions (Supplementary Fig. 5). In order to identify the specific H⁺ transport channels within the bulk of MgO generated by the addition of a heteroatom Al, we investigated the H⁺ diffusion property on the Al–MgO with the Mg/Al composition of five which accommodated maximum amount of Al without phase transformation. In these trials, the evolutions of HD molecules from Ru/MgO and Ru/Al–MgO were monitored by mass spectrometry (MS) under an $H_2$ atmosphere by heating following to $D_2$ reductions as shown in Supplementary Fig. 6. The Ru/Al–MgO generated three Gaussian peaks whereas two Gaussian peaks in the case of the Ru/MgO (Fig. 3b). Only first HD production between 50 and 150 °C can be originated from H⁺ diffusion on the surface, respectively, according to the in situ DRIFT measurements (Supplementary Figs. 7 and 8). Hence, the second and third HD productions from Ru/Al–MgO are based on the specific H⁺ transport channels within the bulk. The area ratios of these peaks to the first peak were calculated to be 5.6 and 5.9, respectively, even though it was only 1.3 in the case of the second peak generated from Ru/MgO, showing that superior H⁺ storage capacity of Al–MgO was originated from the two H⁺ transport channels within the

bulk region. Note that H⁺ diffusion on the Al–MgO occurs at lower temperature than $CeO_2$ and $WO_3$ and therefore the diffusion rate is supposed to be superior to these two reducible metal oxides (Supplementary Fig. 9). To identify specific channels with in the Al–MgO, additional trials were performed as a function of a heating rate ($\beta$) and obtained Kissinger plots for each HD production peak (Supplementary Figs. 10 and 11)[32]. From the slopes and the intercepts of obtained approximated lines, the activation energies ($E_a$) and the pre-exponential factors ($A$) for H⁺ diffusion reactions corresponding to each HD production were respectively calculated (Fig. 3b), based on the following equation[32],

$$\ln \frac{\beta}{T_m^2} = -\frac{E_a}{R}\frac{1}{T_m} + \ln A \qquad (1)$$

where $T_m$ and $R$ are the temperature associated with maximum HD evolution and the universal gas constant, respectively. The $E_a$ values for HD evolutions from Ru/Al–MgO corresponding to peak 1 and peak 2 were calculated to be 37.3 and 41.9 kJ/mol, respectively. The $E_a$ value of peak 2 was only less than 10 kJ/mol larger than that of peak 1, and this trend was the same in the case of Ru/MgO. This result suggests that the H⁺ diffusion sites related to peak 2 were similar to those responsible for peak 1. Therefore, the H⁺ diffusion sites related to peak 2 are likely comprised unsaturated oxygen atoms located in the bulk. It has been reported that some oxygen atoms in the inversion spinel (IS) crystals can become unsaturated[25]. Considering that 19.2% of the Al in the present Al–MgO occupied tetrahedral sites to form IS units, this material would have contained a large amount of unsaturated oxygen atoms which provide H⁺ diffusion sites within its structure. To confirm this issue, we obtained a two-dimensional map based on ¹H-²⁷Al solid-state NMR data acquired from the Ru/Al–MgO after $H_2$ reduction (Fig. 2c). A cross peak was observed at (74.3, 7.0) together with an intense peak at (15.5, 5.5), showing the presence of protonic H near $Al_{Td}$ as well as $Al_{Oh}$[33]. This result suggests the formation of $Al_{Td}$-O-H groups within the Al–MgO[34]. Hence, the unsaturated oxygen atoms evolved by the incorporation of $Al_{Td}$ provide H⁺ transport channels into the bulk region with the Al–MgO. On the other hand, the $E_a$ for peak 3 was calculated to be 178.2 kJ/mol, that was more than 100 kJ/mol larger than that for peak 1, suggesting that pathway associated with peak 3 is less diffusive for H⁺ than that of peaks 1 and 2. Moreover, the value of $A$ for peak 3 was ten orders of magnitude higher than those for peaks 1 and 2, demonstrating that the cross-sections of the H⁺ diffusion sites related to peak 3 were larger than those of unsaturated oxygen atoms because the pre-exponential factor is associated with the collision cross-section of a reactive substrate[35]. Based on the extremely high $E_a$ and $A$ values, the H⁺ transport channel in the bulk of Al–MgO related to peak 3 was assigned to three-dimensionally extended $V_{Cat}$ according to following reasons. First, it is known that atomic hydrogen can be

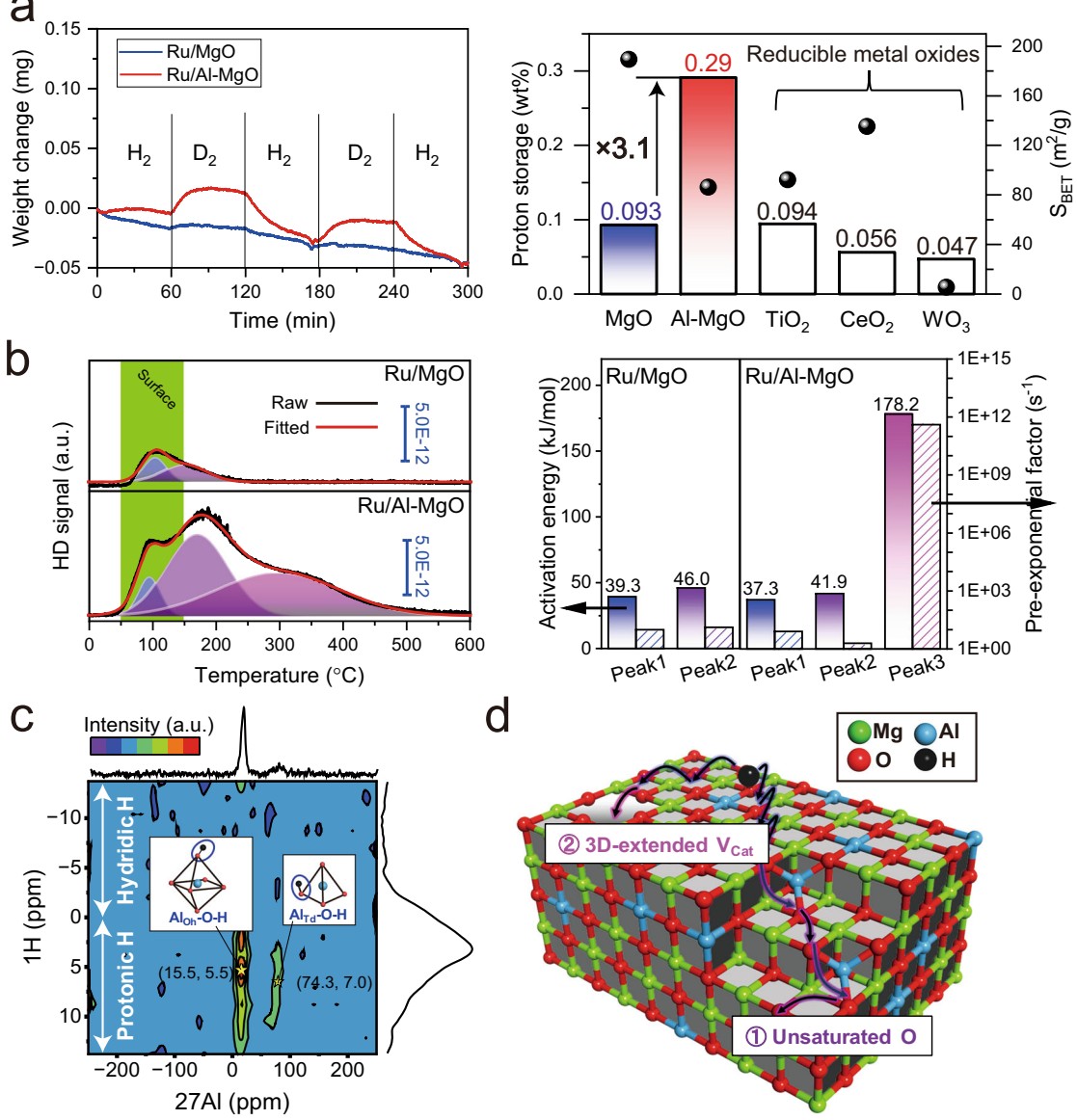

**Fig. 3 | Superior proton diffusion performance of Al–MgO. a** Variations in masses of Ru-loaded MgO and Al-MgO under alternating $H_2$ and $D_2$ atmospheres at 400 °C as determined by TG analysis. Outset: Calculated $H^+$ storage capacities and specific surface areas of the MgO and Al–MgO together with values for reference reducible metal oxides. **b** Data from kinetic analyses of $H^+$ diffusion. Left: HD production during $H_2$-TPD following $D_2$ annealing as obtained by mass spectrometry while heating at a rate of 5 °C min⁻¹. The temperature range where $H^+$ diffusion occurs on the surface region is highlighted as a green shading. Right: activation energies (fully shaded bars) and pre-exponential factors (partly shaded bars) for each HD production trial using Ru-loaded MgO and Al-MgO. **c** A two-dimensional ¹H-²⁷Al HET-COR map obtained for Ru/Al-MgO after $H_2$ reduction. The spectra on the right and upper axes represent the projection of obtained 2D map to ¹H and ²⁷Al axes, respectively. **d** A diagram showing the proposed $H^+$ transport channels within the Al-MgO. Unsaturated oxygens associated with $Al_{Td}$ and 3D-extended $V_{Cat}$ provide $H^+$ transport channels into Al-MgO, respectively.

trapped in vacancies within bulk materials based on stable H-vacancy interactions[36] such that the $E_a$ for diffusion is increased. Second, the octahedral cation sites close to a $V_{Cat}$ have a 43.9% probability of being vacant in Al–MgO (Supplementary Note 1), which significantly increases $A$. It has been reported that $V_{Cat}$ can react with $H_2$ molecules to initiate hydrogen spillover[16]. Our result suggests that they facilitate not only the $H_2$ dissociation step but also the $H^+$ diffusion steps in the hydrogen spillover process. It should be noted that the higher calcination and reduction temperatures resulted in the sluggish of $H^+$ storage capacity because of the decrease of $H^+$ transportation channels within the bulk regions (Supplementary Figs. 12 and 13). From the above results, we can conclude that the Al–MgO was able to store and transport a large amount of $H^+$ as a consequence of the two types of $H^+$

transport channels that were generated within the bulk region of Al–MgO as well as the surface unsaturated oxygen (Fig. 3d). Specifically, adjoining unsaturated oxygen atoms distributed within the bulk transported $H^+$ via the formation of O–H bonds (channel 1) and three-dimensionally extended $V_{Cat}$ within the bulk transported $H^+$ via H-vacancy interactions (channel 2).

## e⁻ diffusion property over Al–MgO

The hydrogen spillover ability of Al–MgO was further assessed in terms of e⁻ diffusion. WO₃ has been employed as a marker of hydrogen spillover in several reports[37,38] because it shows change of its color from yellow to bronze by the acceptance of e⁻ due to the mixing of the valence transfer bands of $W^{6+}$ and $W^{5+}$[39]. Herein, the e⁻ diffusion

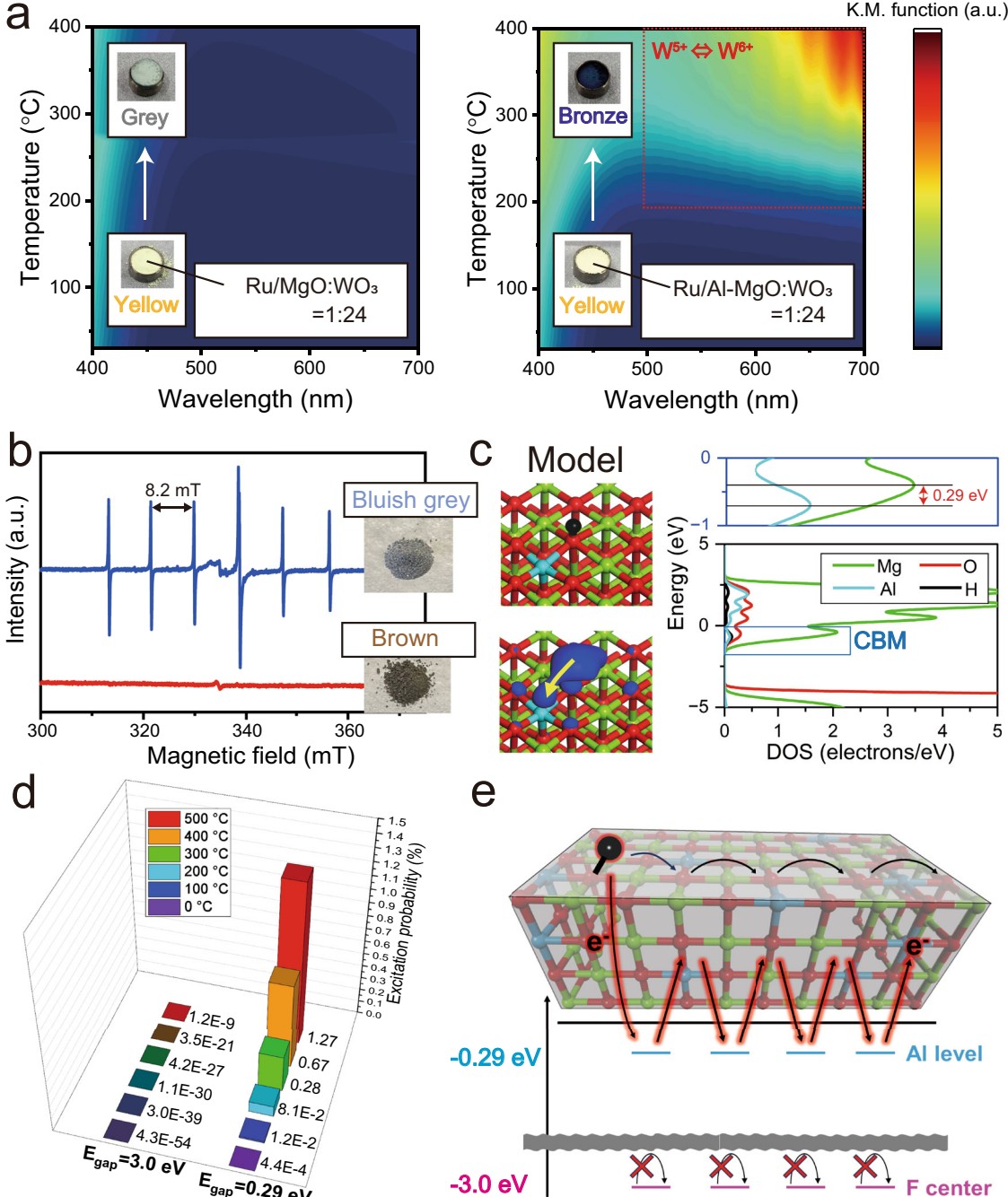

**Fig. 4 | Enhanced electron conduction on Al−MgO. a** Changes in visible light absorption for $WO_3$ mixed with Ru-loaded MgO (left) and Al−MgO (right) based on in situ UV-vis spectroscopy. Inset: photographs of the specimens before and after the trials. **b** ESR spectra acquired from Ru-loaded MgO and Al−MgO after $H_2$ reduction. Inset: photographs of the specimens. **c** Results obtained from DFT calculations for the model in the upper left of this subfigure. A diagram showing the spin electron density for a hydrogen on MgO having one Al atom (bottom left) and the PDOS results (right). The plot in the upper right is an enlarged view of the CBM. **d** Calculated excitation probability values for electrons at various temperatures as determined using an $E_{gap}$ of 3.0 or 0.29 eV. **e** Proposed e⁻ migration pathways within Al−MgO mediated by the donor levels associated with added Al.

associated with hydrogen spillover was studied from the visible light absorption of $WO_3$ mixed with Ru-supported specimens under an $H_2$ reduction atmosphere with in situ UV-vis measurements (Fig. 4a). $WO_3$ mixed with the Ru/MgO showed no visible light absorption up to 400 °C, demonstrating that the $WO_3$ did not accept electrons. Hence, there was evidently minimal e⁻ diffusion on the MgO. Interestingly, $WO_3$ mixed with the Ru/Al−MgO exhibited strong visible light absorption above 200 °C and the bronze coloration of the sample following this experiment confirmed that the $WO_3$ had been reduced[39].

This result demonstrated that the H⁺ diffusion on the Al−MgO was accompanied by e⁻ migration, allowing hydrogen spillover to occur as in the case of reducible metal oxides even though $Mg^{2+}$ and $Al^{3+}$ are the least reducible. Note that H atoms do not migrate directly from Ru to $WO_3$ but through support materials, since there is a clear difference in the visible light absorptions of $WO_3$ mixed with Ru/MgO and Ru/Al−MgO. According to the electron spin resonance (ESR) spectra, only Ru/MgO generated an intense hyperfine pattern consisting of six signals with a hyperfine constant of 8.2 mT ascribed to the formation of F

centers in MgO after $H_2$ reduction (Fig. 4b)[40]. F centers have reported to generate defect levels in MgO at least 3 eV lower than the conduction band minimum (CBM)[41], which inhibited $e^-$ diffusion occurring simultaneously with $H^+$ migration on the MgO. Therefore, the addition of Al prevented the deep trapping of $e^-$ and so enhanced $e^-$ diffusion during $H^+$ diffusion, enabling hydrogen spillover on the non-reducible Al–MgO. The effect of Al doping on $e^-$ diffusion during hydrogen spillover was investigated by performing density functional theory (DFT) calculations based on a MgO (001) facet in which an Mg was substituted by an Al (Fig. 4c). Electron spin density calculations indicated that an $e^-$ derived from an H atom migrated to the Al, suggesting that $e^-$ migration proceeding together with $H^+$ diffusion may be initiated by the donation of $e^-$ from H atoms to Al on the Al–MgO. As a means of better understanding the movement of $e^-$ on this material, the position of the Al level in the band structure of the present model was investigated on the basis of a partial density of states (PDOS) analysis. This level was found to be situated 0.29 eV below the CBM of MgO and even lower that the level of the 1 s orbital of spilled hydrogen (Fig. 4c). A similar trend was observed in the results of additional calculations based on varying the position of Al (Supplementary Fig. 14). According to the Fermi distribution function, an energy gap of 3.0 eV (the estimated minimum energy difference previously reported between the F center defect levels and the CBM of MgO)[41] excites $e^-$ between bands with a probability of $\sim 1.1 \times 10^{-30}$%, whereas the probability is as high as $8.1 \times 10^{-2}$% for an energy gap of 0.29 eV at 200 °C, respectively (Fig. 4d and Supplementary Fig. 15). That is, an Al atom doped into MgO can both accept an $e^-$ from a spilled H atom and donate an $e^-$ to the MgO CBM. This effect promotes hydrogen spillover on Al–MgO even though this compound comprises the two poorly reducible elements Mg and Al. The $e^-$ diffusion characteristics of the MgO and Al–MgO and the origins of this phenomenon are illustrated in Fig. 4e. In the case of the pristine MgO, $H^+$ and $e^-$ form F centers and produce defect levels at least 3.0 eV below the CBM of MgO that capture $e^-$. Thus, $e^-$ conduction is inhibited and only $H^+$ diffuses over the MgO. However, the Al heteroatom provides donor levels at 0.29 eV below the MgO CBM that not only inhibits deep trapping of $e^-$ into F centers but also promotes $e^-$ migration with only a minimal thermal energy input. Therefore, hydrogen spillover based on the concurrent diffusion of $H^+$ and $e^-$ can occurs even though the cations in this material are not readily reduced.

## Hydrogen spillover effect on $CO_2$ hydrogenation

The reactivity of hydrogen spillover on Al–MgO was evaluated based on the catalytic $CO_2$ hydrogenation into CO and $CH_4$ between 300 and 500 °C (Fig. 5a). In these trials, 0.1 wt%-Pt supported Al–MgO (Pt/Al–MgO) and 3.0 wt%-Ni supported Al–MgO (Ni/Al–MgO) catalysts were physically mixed at a mass ratio of 1:1, referred to herein as (Pt + Ni)/Al–MgO. In this catalyst, the Pt, Al–MgO, and Ni promoted the dissociation of $H_2$ molecules, the transportation of atomic hydrogen from Pt to Ni sites, and the utilization of atomic hydrogen for the $CO_2$ hydrogenation reaction, respectively. No significant phase transformation occurred during the deposition of Pt and Ni and the subsequent $H_2$ reduction and the mean diameters of Pt and Ni on Al–MgO were 9.26 and 49.3 nm, respectively (Supplementary Fig. 16). The major hydrogenated product from separated Pt/Al–MgO and Ni/Al–MgO was CO and each $CO_2$ conversion moderately increased from 350 to 400 °C, giving 2.5% and 4.3% at 400 °C, respectively (Fig. 5b). In contrast, the (Pt + Ni)/Al–MgO exhibited a significant increase in $CO_2$ conversion in the same temperature range from 350 to 400 °C. At 400 °C, this material provided a $CO_2$ conversion 4.6 times higher than the mean values obtained from the same amount of Pt/Al–MgO and Ni/Al–MgO, as shown in Fig. 5b. The (Pt + Ni)/Al–MgO also demonstrated product selectivity that was different from those of the individual catalysts, giving 8.2% $CH_4$ and 7.6% CO at 400 °C. The evolution of one mole of $CH_4$ requires four moles $H_2$, whereas the production of one

mole of CO requires only one mole $H_2$. Based on this, the $H_2$ utilization efficiencies, the proportion of reacted $H_2$, of each catalyst during this reaction were calculated (Fig. 5b). The (Pt + Ni)/Al–MgO exhibited an $H_2$ utilization efficiency 11.4 times greater than the mean value for the Pt/Al–MgO and Ni/Al–MgO, demonstrating that a greater quantity of $H_2$ was utilized for $CO_2$ hydrogenation over the (Pt + Ni)/Al–MgO owing to the hydrogen spillover effect provided by the Al–MgO. We performed the same catalytic trials using various specimens with different designs: catalysts which employed Ru as the dissociation site of $H_2$ molecules (Supplementary Fig. 17), a catalyst which co-supported Pt and Ni (Supplementary Fig. 18), catalysts utilizing the Al–MgO calcined at different temperature as support materials (Supplementary Figs. 19 and 20), and catalysts with MgO as support material (Supplementary Fig. 21). Although the catalytic performances of these catalysts were superior to those of the individual catalysts respectively, their activity enhancements were moderate compared with that of (Pt + Ni)/Al–MgO shown in Fig. 5b. It has been reported that the $CO_2$ hydrogenation reaction on Ni was initiated by the dissociation of $CO_2$ to *CO and *O species and evolved surface oxygen passivated the Ni surface and lowered its activity[42]. Hence, it is apparent that the Ni surfaces poisoned by oxygen atoms were readily reduced by the spilled hydrogen on Al–MgO. In order to confirm this, the redox behaviors of Ni species deposited on Ni/Al–MgO, (Pt + Ni)/MgO, and (Pt + Ni)/Al–MgO were examined from the linear fitting processing with Ni and NiO for the Ni K edge in situ X-ray absorption near edge structure (XANES) under alternating $H_2$ and $O_2$ atmospheres at 350 °C, respectively (Fig. 5c). As the results, the fraction of $Ni^0$ calculated from $Ni^0/(Ni^0 + Ni^{2+})$ was 0.35 for (Pt + Ni)/Al–MgO, which is larger by 0.22 than Ni/Al–MgO. This suggests that the reduction of Ni species is promoted by the presence of Pt under $H_2$ reduction atmosphere. Additionally, this value was larger by 0.18 than that of (Pt + Ni)/MgO. This tendency was repeatedly obtained in the second $H_2$ dosage following to $O_2$ dosage. These results demonstrate that hydrogen spillover on Al–MgO significantly endows the reduction of Ni species and its reduction performance is superior to $H^+$ diffusion on MgO. The results of $H_2$-TPR measurements also support that the promotional effect of hydrogen spillover in the reduction of Ni species (Supplementary Fig. 22). Hence, it can be concluded that the hydrogen spillover on Al–MgO immediately removed oxygen atoms poisoning Ni surface during $CO_2$ hydrogenation atmosphere that significantly enhanced the catalysis of Ni (Fig. 5d).

## Discussion

The hydrogen spillover was achieved on a non-reducible MgO by a heteroatom Al doping. The hydrogen spillover on the Al–MgO was disentangled from the viewpoint of $H^+$ diffusion and $e^-$ diffusion and utilized for the catalysis. The Al–MgO allowed hydrogen spillover that is the coupled $H^+$ and $e^-$ diffusion and exhibited superior $H^+$ storage capacity to typical reducible metal oxides even though it comprises only Earth-abundant elements. Compared with pristine MgO and typical reducible metal oxides composed of rare elements, the $H^+$ capacity of Al–MgO was increased by a factor in excess of 3.1. A combination of kinetic and spectroscopic assessments provided evidence that the addition of Al generated both unsaturated oxygen sites and $V_{Cat}$. These provided transport channels into the MgO bulk and significantly enhanced the proton storage capacity of the material. In situ UV-vis spectra confirmed that electron conduction occurred on the Al–MgO in conjunction with proton migration in spite of the poor reducibility of the components of this material. Computational analyses showed that the Al impurity level introduced below the MgO CBM promoted the movement of electrons between these two levels via thermal excitation. Hydrogen spillover on the Al–MgO was also found to accelerate the catalytic hydrogenation of $CO_2$ by removing surface oxygen on Ni sites that would otherwise poison the catalyst. This effect resulted in a 11.4-fold increase in hydrogen utilization

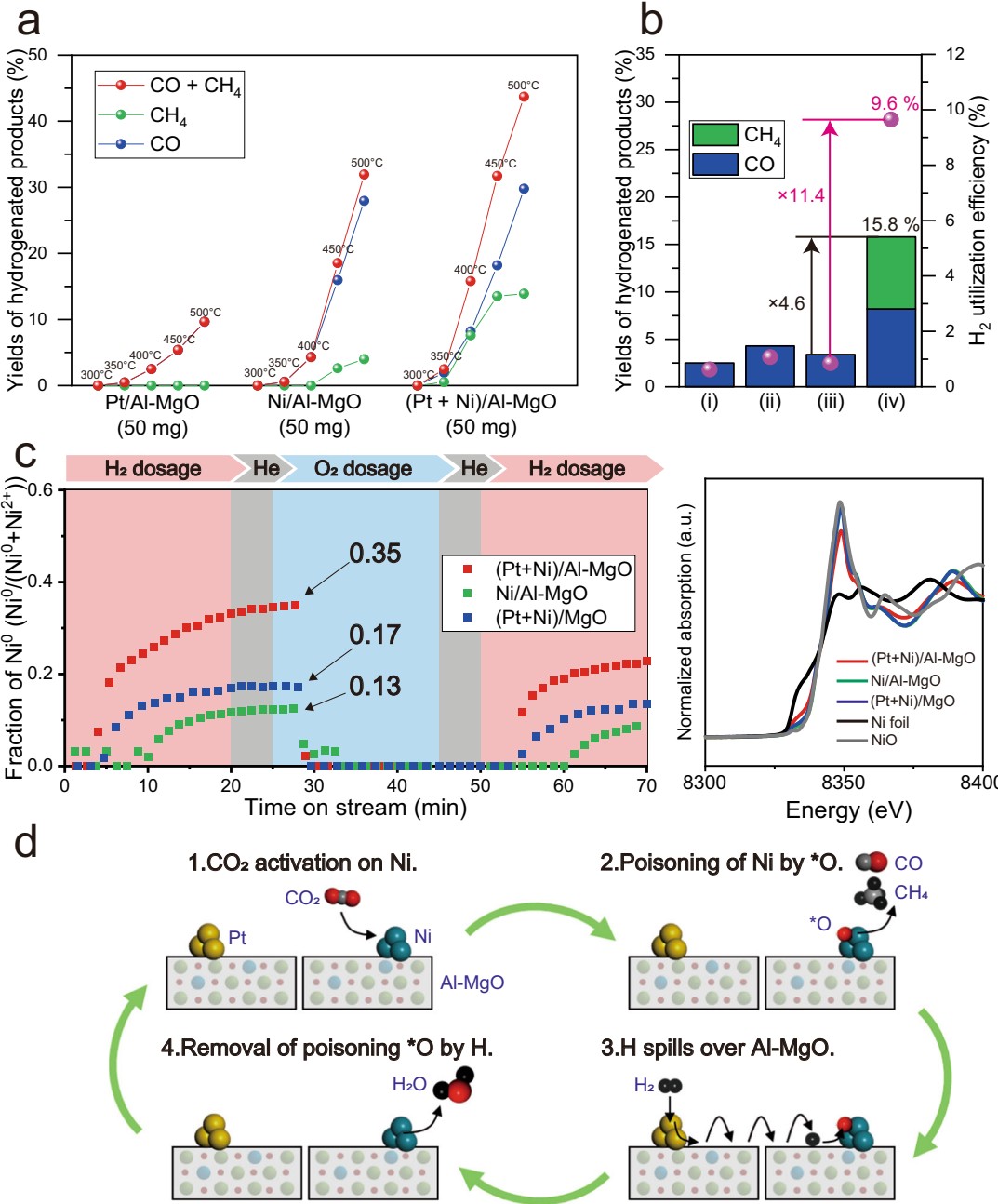

**Fig. 5 | Enhanced catalytic CO₂ hydrogenation by hydrogen spillover on Al-MgO. a** Yields of hydrogenated products obtained from Al–MgO catalysts loaded with Pt, Ni, or (Pt + Ni) at various temperatures. **b** Bars: catalytic CO₂ conversion to CO and CH₄ at 400 °C for (i) 50 mg of Pt/Al–MgO, (ii) 50 mg of Ni/Al–MgO, (iii) average of values for Pt/Al–MgO and Ni/Al–MgO, and (iv) 50 mg of (Pt + Ni)/Al–MgO. Plot: H₂ utilization efficiencies during the CO₂ hydrogenation reaction. **c** Left: The fraction of metallic Ni⁰ species in the (Pt + Ni)/Al–MgO, Ni/Al–MgO, and (Pt + Ni)/MgO based on linear fitting for Ni K edge XANES spectra obtained while switching between H₂ and O₂ atmospheres at 350 °C. Right: The Ni K edge XANES spectra for each plot pointed in the left figure together with Ni and NiO. **d** A schematic diagram illustrating how hydrogen spillover on Al–MgO promotes the catalysis of Ni in CO₂ hydrogenation.

efficiency. The material demonstrated herein could lead to the development of novel hydrogen handling technologies as a means of achieving a sustainable, hydrogen-based society.

## Methods
### Materials
RuCl₃·$n$H₂O (≥99.9% pure), NiCl₂·6H₂O (≥98% pure), H₂PtCl₆·6H₂O (≥98.5% pure), Mg(NO₃)₂·6H₂O (≥99.0% pure), Al(NO₃)₃·9H₂O (≥98.0% pure), Na₂CO₃ (≥99.8% pure), NaOH (≥97.0% pure), poly(N-vinyl-2-pyrrolidone) (average molecule weight 40,000), ethylene glycol (EG, ≥99.0% pure) and ethanol (≥99.5% pure) were supplied by Nacalai

Tesque. WO₃ (99.5% pure) was obtained from Wako Pure Chemical Industries, Ltd. All commercially available chemicals were used as received.

### Synthetic procedures
The Al–MgO (Mg/Al = 5) specimen was synthesized using a conventional co-precipitation method reported previously[43]. In this process, Mg(NO₃)₂·6H₂O (12.820 g, 0.05 mol) and Al(NO₃)₃·9H₂O (3.751 g, 0.01 mol) were first dissolved in distilled water (100 mL). This solution was then slowly added to 60 mL of an aqueous solution of Na₂CO₃ (3.180 g, 0.03 mol) and NaOH (2.800 g, 0.07 mol) at 65 °C with stirring.

The resulting mixture was vigorously stirred for a further 18 h after which the precipitate was divided into four lots, each of which was combined with 45 mL distilled water and centrifuged five times. The white slurry obtained from this process was dried for at least 6 h at 110 °C, following which the resulting solid was ground into a powder for 3 min and then calcined at 400 °C under air for 1.25 h. Pure MgO and x-Al–MgO (where x is the Mg/Al ratio) specimens were prepared in the same manner by omitting or varying the amount of $Al(NO_3)_3$·$9H_2O$.

Ru/Al–MgO (Mg/Al = 5) was synthesized using a conventional impregnation method followed by a standard hydrogen reduction procedure. Specifically, $RuCl_3$·$nH_2O$ (0.0270 g) was added to a mixture of Al–MgO (0.5 g) and distilled water (100 mL). The resulting suspension was stirred at room temperature for more than 1 h, after which the water was evaporated by heating the material at 60 °C under vacuum. The powder obtained from this process was reduced by heating to 400 °C at a rate of 5 °C/min under a 20 mL/min flow of $H_2$ and then holding the specimen at that temperature for 2 h to yield Ru/Al–MgO containing 2.0 wt% Ru. MgO samples loaded with Ru were prepared in the same manner.

A colloidal Pt suspension was produced using a conventional polyol method as previously reported[44]. In this process, $H_2PtCl_6$·$6H_2O$ (1.0 g) was added to EG (50 mL) with stirring while NaOH (1.0 g) was dissolved in EG (50 mL) to produce a second solution. These two solutions were then combined and stirred for 10 min under a 50 mL/min flow of Ar at room temperature. The resulting mixture was stirred at 160 °C under a continuous flow of Ar for 3 h to obtain a suspension containing 0.377 g colloidal Pt/100 mL. A $NiCl_2$·$6H_2O$ solution (0.1 M) was also prepared by dissolving $NiCl_2$·$6H_2O$ (0.238 g) in distilled water (10 mL), after which Pt/Al–MgO and Ni/Al–MgO were synthesized via a typical impregnation method. In this synthesis, either the colloidal Pt suspension (0.796 mL) or the $NiCl_2$·$6H_2O$ solution (1.53 mL) described above was added to a mixture of Al–MgO (0.3 g) and distilled water (100 mL). Each suspension was stirred at room temperature for more than 1 h after which the water was evaporated by heating at 60 °C under vacuum to give Pt/Al–MgO (0.1 wt% Pt) or Ni/Al–MgO (3.0 wt% Ni).

(Pt + Ni)/Al–MgO was prepared by combining Pt/Al–MgO and Ni/Al–MgO at a mass ratio of 1:1 and mixing for 3 min.

## Characterizations

Powder XRD patterns were acquired using a Rigaku Ultima IV diffractometer with Cu Kα radiation ($\lambda$ = 1.54056 Å). High-resolution TEM (HR-TEM) images and associating electron diffraction (ED) patterns were obtained using a JEM-ARF200F instrument (JEOL Ltd.) and were analyzed using the ReciPro software package[45]. Nitrogen adsorption-desorption isotherms were acquired at −196 °C using a BELSORP-max system (MicrotracBEL Corp.). Samples were degassed at 120 °C for 3 h under vacuum to vaporize physisorbed water prior to each trial. The specific surface areas of the synthesized oxides were calculated by the Brunauer–Emmett–Teller (BET) method using nitrogen adsorption data. Thermogravimetry (TG) data were obtained with a Rigaku Thermo plus EVO2 TG8121 system under sequential 100 mL/min flows of 5.0% $H_2$ and 5.0% $D_2$ diluted with $N_2$ gas at 400 °C for 10 mg of Ru-supported specimens. The quantity of protons stored by the Ru supported on each sample was calculated from the equilibrium difference in $H_2$ and $D_2$ masses. In situ diffuse reflectance infrared Fourier transform spectroscopy (DRIFT) analyses were conducted using an IR Spirit instrument (Shimadzu) equipped with a heating chamber and connected to a gas-exchange system. During these assessments, each sample was heated at 50, 150, or 250 °C under sequential flows of $H_2$ and $D_2$, and spectra were acquired 10 min after exposure to $D_2$ at each temperature. Baseline correction was applied to each spectrum except between 2600 and 2800 cm$^{-1}$ (corresponding to the region associated with O–D stretching vibrations). The generation of HD molecules ($m/z$ = 3) via the H$^+$–D$^+$ exchange process during $H_2$-TPD followed by annealing under $D_2$ was monitored using MS with a BELMass spectrometer connected to a BEL-CAT instrument. During each experiment, the Ru-loaded specimen was annealed under a $D_2$ flow at 400 °C for 1 h as a pre-treatment. Following this, $H_2$-TPD was performed from 0 to 600 °C while applying a heating rate of 2, 5, or 10 °C/min and monitoring HD production by MS. The resulting data were deconvoluted to provide multiple Gaussian peaks. In situ ultraviolet-visible adsorption spectroscopy (UV-vis) analyses were conducted using a V-750 spectrometer (JACSO International Co., Ltd.). Prior to each trial, the Ru-loaded specimen was combined with $WO_3$ at a mass ratio of 1:24 to produce the test specimen. During each analysis, changes in the visible light absorption of the $WO_3$ under hydrogen reduction conditions were monitored from room temperature to 400 °C. Electron spin resonance (ESR) data were obtained from the Ru-loaded specimens at room temperature using a JEOL RESONANCE JES X320 spectrometer. Solid-state magic-angle spinning nuclear magnetic resonance spectra of Ru/Al–MgO specimens were obtained using a JEOL RESONANCE ECA 400WB spectrometer operating at 9.4 T with a spinning rate of 6.5 kHz. An 8.0 mm JEOL HXMAS probe with a resonance frequency of 104.2 MHz was employed in all cases. The $^{27}$Al chemical shifts were determined relative to that of gibbsite ($Al(OH)_3$)[46]. $H_2$-temperature programmed reduction analyses were performed using a BEL-CAT instrument by heating 20 mg specimens at 5 °C/min from 50 to 600 °C under a 5.0% $H_2$/Ar flow. Ni K-edge in situ X-ray absorption fine structure (XAFS) data were acquired in the transmission mode using the 01B1 beamline station at the SPring-8 facility operated by JASRI in Harima, Japan (proposal no. 2022B1807) in conjunction with a Si (111) monochromator. As a pre-treatment, each pelletized sample was placed in a batch-type in situ XAFS cell and heated at 10 °C/min from 50 to 500 °C and then held at that temperature for 30 min under a 5.0% $H_2$/He flow. Subsequently, 5.0% $O_2$/He and 5.0% $H_2$/He was alternately introduced into the XAFS cell which was heated at 350 °C for 20 min, during which time XAFS spectra were obtained at 1 min intervals. The XAFS data were processed using the ATHENA program (Demeter).

## Catalytic trials

The catalytic performance of each material was evaluated using a fixed-bed reactor system in which 50 mg of catalyst (Pt/Al–MgO or Ni/Al–MgO or Pt-Ni/Al–MgO) was held in a quartz cell having an internal diameter of 17 mm within an electric oven. Each as-synthesized catalyst specimen was pretreated by heating to 500 °C at a rate of 5 °C/min in a flow of $H_2$ (20 mL/min) for 2 h. The sample was subsequently exposed to a $N_2$/$H_2$/$CO_2$ mixture having a 5/4/1 molar composition (total flow of 50 mL/min, space velocity = 30,000 mL/g/h). The reaction products generated at 300, 350, 400, 450, and 500 °C were analyzed online using a gas chromatograph (Shimadzu GC-14B) and employing an activated carbon column connected to a thermal conductivity detector followed by a flame ionization detector equipped with a methanizer. Each catalyst was held at the target temperature for 10 min before checking the catalytic performance.

## Computational methods

PDOS data were calculated using DFT with periodic boundary conditions employing the CASTEP plane-wave-based program[47,48]. The generalized gradient approximation exchange-correlation functional proposed by Perdew, Burke, and Ernzerhof (PBE) was used together with ultrasoft-core potentials[49]. The basis-set cutoff energy was set to 340 eV and the electronic configurations of the H, O, Mg, and Al atoms were $1s^1$, $2s^2 2p^4$, $2p^6 3s^2$, and $3s^2 3p^1$, respectively. A MgO (001) slab model was generated by cleavage from a bulk crystalline structure using lattice parameters of $a$ = 8.9333, $b$ = 8.9333 and $c$ = 19.2112 and lattice angles of $\alpha = \beta = \gamma = 90°$. Each slab model contained a vacuum region with a thickness of 15 Å in the $c$-direction. Each PDOS calculation and the associated mapping were performed following the geometric optimization of the computational model during which all atoms were

relaxed. The total spin density was mapped for each optimized computational model using an isosurface value of 0.015 $e$/Bohr$^3$.

## Data availability

All data generated during this study are included in this article and its Supplementary Information or are available from the corresponding authors upon request.

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

## Acknowledgements

The present work was supported by the Kakenhi Grant-in-Aid for Transformative Research Areas(B) (No. 21B206). K.S. thanks JSPS for a Research Fellowship for Young Scientists (no. 23KJ1506). The solid-state MAS-NMR measurements were performed by the Analytical Instrument Facility, Graduate School of Science, Osaka University. TEM experiments were carried out by using a facility in the Research Center for Ultra-High Voltage Electron Microscopy, Osaka University. We thank Prof. Hisayoshi Kobayashi at the Kyoto Institute of Technology for kind support in DFT calculations. The synchrotron radiation experiments for XAFS measurements were performed at the BL01B1 beamline in SPring-8 with approval from JASRI (2023B1805).

## Author contributions

K. S. performed the catalyst preparation, calculation, and characterization, and wrote the manuscript. K. M. supervised all of the project. T. K. helped the catalyst preparation and performed catalytic activity trials. S. I. obtained HR-TEM images. H. Y. helped supervise the project. The manuscript was written through the discussion with all authors. All authors have given approval to the final version of manuscript.

## Competing interests

The authors declare no competing interests.
