## [Peer Review File · Nature Communications]

Heteroatom Doping Enables Hydrogen Spillover via H⁺/e⁻ Diffusion Pathways on a Non-reducible Metal OxideREVIEWER COMMENTS

Reviewer #1 (Remarks to the Author):

The present work demonstrates that MgO containing a moderate amount of Al (Al-MgO) exhibits improved hydrogen spillover performance than pristine MgO. Hydrogen spillover on Al-MgO enhances its H storage capacity and the catalytic performance of Ni during CO₂ hydrogenation reactions. The diffusions of proton and electron on Al-MgO were studied by systematic characterizations and calculations. This paper is interesting and can be considered for publication after major revision. The detailed comments are as follows.

1. The authors claimed that Al-MgO showed outstanding H storage capacity compared with reducible metal oxides (Fig 3a). And they also claimed that "the design of materials based on such common elements that also provide significant hydrogen spillover ability will enable the development of innovative and sustainable technologies for a next-generation hydrogen society". Therefore, it needs to be proven whether Al-MgO is superior to the reducible oxides. The unanswered question in this paper is if the rate and distance of hydrogen spillover on Al-MgO are competitive with those on the reducible oxide, such as TiO₂.
2. The calculation process of H storage capacity and the essential characterizations of reducible metal oxides are missing. How to attribute the improved storage capacity to hydrogen spillover rather than other factors (such as particle size, or metal-support interaction)? Is the storage capacity related to the preparation process of the sample?
3. It is odd that Ru-based catalysts were used to show the improved hydrogen spillover performance, while Pt and Ni-based catalysts were used to show the improved catalytic performance. How about the CO₂ hydrogenation performance of Ru-based catalysts? And the essential characterizations of Pt/Ni-based catalysts, such as HRTEM, XRD, H-D exchange, and H₂-TPD, should be provided.
4. The influence of Al content in Al-MgO sample on the hydrogen spillover effect should be provided.
5. In Fig S8, it seems that at low temperatures (from 50 to 150 oC), hydrogen spillover on Al-MgO is not stronger than on MgO. Unlike that, hydrogen spillover on TiO₂ is also superior at room temperature. Does this limit its potential applications at low temperatures?
6. It is better to provide a schematic diagram that includes both proton and electron migration paths. Does the motion of proton and electron in Al-MgO is coherent (because of charge balance)?
7. Al-MgO was calcined at 400 oC under air for 1.25 h. The H-D experiment was carried out at 400 oC. And the CO₂ hydrogenation was carried out above 300 oC. The influence of calcination temperature of Al-MgO on hydrogen spillover and the catalytic performance should be added.
8. As to the WO₃ color change experiment, the H atoms may not migrate through the MgO/Al-MgO support, they could migrate from Ru particles to WO₃ directly. Is it a powerful characterization tool for electron diffusion?
9. Is it accurate to calculate the average oxygen concentration within the Al-MgO from EDX line analysis results?
10. In Fig. 5c, the quantitative analysis results of in situ XANES, such as linear fitting, should be provided.

Reviewer #2 (Remarks to the Author):

The present report illustrates that the proper introduction of cation vacancies in MgO by Al doping enables an efficient hydrogen spillover from metal sites to the non-reducible MgO support, which is otherwise thermodynamically unfavored. Although I read the manuscript with great interest and would like to recommend its publication in Nature Communications, I am afraid that the authors did not demonstrate the novelty of their findings adequately. My main concerns are addressed as follows.

1. The authors highlighted in the abstract and introduction sections that hydrogen spillover is limited to occur over reducible metal oxides composed of rare elements. In the view of element abundancy, graphitic structures and zeolites have already been reported for their application in hydrogen spillover (e.g., Chem. Rev. 2012, 112, 2714-2738; Nat. Commun. 2014, 5, 3370; Angew. Chem. Int. Ed. 2019, 58, 7668-7672). In addition, the ideas about introducing vacancy sites or semiconductors to promote hydrogen spillover have been comprehensively discussed by Roel Prins in his remarkable review (Chem. Rev. 2012, 112, 2714-2738). I am afraid the authors need to make it clear whether their findings are compatible with the existing knowledge or reflective of new strategies.

2. The authors also emphasized that the Al-MgO sample they made exhibited superior hydrogen spillover performance. However, I am not sure how to define the superior performance. For instance, the authors did not compare Al-MgO with reducible metal oxides (e.g., CeO₂ and TiO₂) in hydrogen spillover.

3. It is noticeable that the authors assessed the proton and electron diffusion properties of the Al-MgO sample separately, while the hydrogen spillover requires atomic H species migrate along solid surfaces via coherent proton-electron movements. In particular, Roel Prins has clearly elucidated that the H⁺ diffusion itself does not correlate with hydrogen spillover (Chem. Rev. 2012, 112, 2714-2738), because H⁺ diffusion can occur readily with H⁺ exchange between surface hydroxyl groups. Therefore, I don't think the content about the proton diffusion is very helpful.

4. Related with the previous issue, the authors discussed the H⁺ transport channel in the bulk of Al-MgO. However, hydrogen spillover is well known as a surface phenomenon. It is suggested to focus more on the surface properties, such as surface compositions and the concentration of surface vacancy sites.

5. In my view, one of the most convincing pieces of evidence about the occurrence of hydrogen spillover on oxide-supported metal catalysts is the reversible formation of surface hydroxy species when the catalyst is exposed to H₂ at elevated temperatures (Nat. Catal. 2013, 6, 710-719). Have the authors observed such phenomena in their DRIFT studies on Ru/Al-MgO and Ru/MgO?

6. About the part of catalytic CO₂ hydrogenation, the authors are strongly suggested to compare the catalytic performance at a constant temperature instead of using the transient experiments shown in Figure 5a, because the steady-state performance is more importance to catalysis. Moreover, what result will be obtained if Pt and Ni are co-loaded on the same Al-MgO support for catalytic CO₂ hydrogenation?

Reviewer #3 (Remarks to the Author):

Kazuki Shun and co-workers reported that non-reducible MgO containing a moderate amount of Al (Al-MgO), comprising Earth-abundant elements, exhibits superior hydrogen spillover performance. The diffusion of H⁺ and e⁻ within Al-MgO were studied by systematic characterizations and calculations. However, the work lacks key experimental results to support their conclusion.

1page3_line64-66 the authors mentioned that the Mg/Al ratio of 5 is used as the optimal ratio. However, you did not explain the relationship between hydrogen spillover and the ratio of Mg/Al. Please provide more experiment results to support the Mg/Al ratio of 5 as the optimal ratio or give a trend of hydrogen spillover performance with different Mg/Al ratios.

2 page 3_line 76, authors used EDX to analyze the content of Al. Please use inductively coupled plasma atomic emission spectroscopy (ICP-AES) to give Al and Mg content. Without accurate content, it is impossible to prove your conclusion including the proportion of the cation vacancies.

3 Fig. S3. I cannot accept analyzing TEM data like this. Please improve the quality of your TEM data, especially the analysis process. Please get FFT pattern transformed from your image and then an inverse Fourier image transformed from FFT pattern by using different pairs of lattice points can be used to explain your results.

In addition, please provide the TEM data of MgO.

4 Fig. S3 and S4 provided almost the same information. Please explain the reason. For EDX mapping, please provide the image of Mg+O+Al. From your results, Al seems more concentrated in some places.

5 Fig. S6. Please explain why the authors used the TEM image for Ru/ MgO but the HAADF-STEM image for Ru/Al-MgO.

6 Fig. S6. Please explain how to identify which particles are Ru particles in the HAADF-STEM image.

For the CO₂ hydrogenation part, I am confused about the experimental design of the authors.

7 page 7 lines 204-205 Authors mentioned that Hence, it is apparent that the Ni surfaces poisoned

by oxygen atoms were readily reduced by the spilled hydrogen on Al-MgO.

Two samples (Pt+Ni)/Al-MgO (i) and Ni/Al-MgO (ii) are under CO₂ hydrogenation condition to check X-ray absorption near edge structure of Ni species. The change of oxidized Ni (ii) and oxidized Ni (i) should be provided.

8 page 7 lines 214 and 215

Authors should provide results of the control sample (Pt+Ni)/MgO under alternating H₂ and O₂ atmospheres to support your conclusion.

We would like to thank all reviewer for the careful review and the valuable comments, which allowed us to improve the paper. Below we list the changes we have made in light of the reviewer's comments.

Reviewer #1 (Remarks to the Author):

The present work demonstrates that MgO containing a moderate amount of Al (Al-MgO) exhibits improved hydrogen spillover performance than pristine MgO. Hydrogen spillover on Al-MgO enhances its H storage capacity and the catalytic performance of Ni during CO₂ hydrogenation reactions. The diffusions of proton and electron on Al-MgO were studied by systematic characterizations and calculations. This paper is interesting and can be considered for publication after major revision. The detailed comments are as follows.

Answer: We would like to thank reviewer 1 for the careful review and appreciate positive comments.

1. The authors claimed that Al-MgO showed outstanding H storage capacity compared with reducible metal oxides (Fig 3a). And they also claimed that “the design of materials based on such common elements that also provide significant hydrogen spillover ability will enable the development of innovative and sustainable technologies for a next-generation hydrogen society”. Therefore, it needs to be proven whether Al-MgO is superior to the reducible oxides. The unanswered question in this paper is if the rate and distance of hydrogen spillover on Al-MgO are competitive with those on the reducible oxide, such as TiO₂.

Answer: Thank you very much for your constructive comment.

We have made two major revisions.

Firstly, we have changed several sentences.

In this paper, one of the most important points we wanted to make is that the **H⁺ storage capacity** of Al-MgO derived from hydrogen spillover was superior to typical reducible metal oxides. However, some sentences you pointed were inappropriate to clarify the above issue because they can be taken to mean not H⁺ storage capacity but comprehensive hydrogen spillover ability of Al-MgO (e.g. rate and distance) is superior.

In order to avoid confusion, we modified the corresponding sentences as follows.

Old [1-1]: The present work demonstrates that non-reducible MgO containing a moderate amount of Al (Al-MgO), comprising Earth-abundant elements, exhibits superior hydrogen spillover performance. This material has a H storage capacity 3.1 times greater than those of various standard metal oxides based on hydrogen transport channels within its bulk region.

New [1-1]: The present work demonstrates that a non-reducible MgO with heteroatom Al dopants (Al-MgO) allows hydrogen spillover in the same way as reducible metal oxides. Furthermore, a H⁺ storage capacity of this materials owing to hydrogen spillover was 3.1 times greater than those of various standard metal oxides based on H⁺ transport channels within its bulk region.

Old [1-2]: The design of materials based on such common elements that also provide significant hydrogen spillover ability will enable the development of innovative and sustainable technologies for a next-generation hydrogen society.

New [1-2]: The design of materials based on such common elements that also allow hydrogen spillover ability will enable the development of innovative and sustainable technologies for a next-generation hydrogen society.

Old [1-3]: The present work demonstrates that Al-doped MgO (Al-MgO), which contains only Earth-abundant elements, exhibits excellent hydrogen spillover performance.

New [1-3]: The present work demonstrates that a non-reducible Al-doped MgO (Al-MgO), which contains only Earth-abundant elements, allows hydrogen spillover in the same way as reducible metal oxides.

Old [1-4]: The non-reducible Al-MgO exhibited superior hydrogen spillover ability despite being composed of only Earth-abundant elements.

New [1-4]: The Al-MgO allowed hydrogen spillover that is the coupled H⁺ and e⁻ diffusion and exhibited superior H⁺ storage capacity to typical reducible metal oxides even though it comprises only Earth-abundant elements.

Secondly, we have added a supplemental figure and relevant sentence to provide data concerning

hydrogen spillover rate.

We previously reported that hydrogen spillover occurred around 50, 150, and 250 °C on the Ru-supported TiO₂, CeO₂, and WO₃ by observing O–D bonds derived from D⁺ diffusion by *in situ* DRIFT measurements. In this regard, the Ru/Al-MgO exhibited hydrogen spillover from 50 to 150 °C. This occurrence temperature is related to the activation energy, which is a function of reaction rate (*Chem. Sci.*, 2022, 13, 8137-8147). Therefore, we suppose that the hydrogen spillover rate on the Al-MgO is superior to CeO₂ and WO₃ but inferior to TiO₂.

In order to clarify above issue, we added *in situ* DRIFT measurements for Ru-supported TiO₂, CeO₂ and WO₃ and a relevant sentence as follows.

New [1-5]: The area ratios of these peaks to the first peak were calculated to be 5.6 and 5.9, respectively, even though it was only 1.3 in the case of the second peak generated from Ru/MgO, showing that superior H⁺ storage capacity of Al-MgO was originated from the two H⁺ transport channels within the bulk region. Note that H⁺ diffusion on the Al-MgO occurs at lower temperature than CeO₂ and WO₃ and therefore the diffusion rate is supposed to be superior to these two reducible metal oxides (Supplementary Fig. 9). To identify specific channels in the Al-MgO...

In terms of distance, we estimate that spilled hydrogen on the Al-MgO can migrate longer than on CeO₂. Al-MgO allowed the interparticle hydrogen spillover to improve the catalytic activity of Ni (Fig. 5). It has been previously reported that the interparticle hydrogen spillover on CeO₂ is inhibited by the CeO₂-CeO₂ interface and occur after the sintering of CeO₂ particles (*JACS Au*, 2023, 3.,8, 2299-2313). We are also interested in the difference in hydrogen spillover distance between the Al-MgO and reducible metal oxides, but we would like to investigate it in the next report because it will take a lot of time to establish the precise method and it would be deserved as a one advanced paper as reported (*Nature*, 2017, 541, 68-71. *J. Am. Chem. Soc.*, 2023, 145, 3, 1631-1637. *ASC Nano*, 2023, 17, 2, 1091-1099.).

2. The calculation process of H storage capacity and the essential characterizations of reducible metal oxides are missing. How to attribute the improved storage capacity to hydrogen spillover rather than other factors (such as particle size, or metal-support interaction)? Is the storage capacity related to the preparation process of the sample?

Answer: Thank you very much for your useful comment.

We investigated the influence of two important factors, such as the calcination temperature and the reduction temperature in the preparation process, on the H^+ storage capacity.

As the calcination temperature of the Al-MgO increased from 400 °C, the H^+ storage capacity drastically decreased. This result demonstrates that the Ru supported Al-MgO calcinated at 400 °C, which is provided in **Fig. 3 (a)**, shows higher H^+ storage capacity than those calcinated at 600 °C and 800 °C. From the evolution of HD from Ru supported specimens under a H_2 atmosphere following to D_2 reduction at elevated temperature, the H^+ storage capacity ratio of the bulk regions to the surface regions was calculated to be 6.7 and 1.5 for the Al-MgO calcined at 600 and 800 °C, respectively, even though that of Al-MgO calcinated at 400 °C was as large as 11.5. Therefore, the decrease in H^+ storage capacity with calcination temperature is attributable to the decrease in H^+ transportation channels, such as cation vacancies (V_{Cat}) and tetrahedral-coordinated Al (Al_{Td}) within the bulk of Al-MgO. According to the phase diagram, MgO and Al_2O_3 are thermodynamically difficult to form solid

solution as below (*Ceramics – Silikaty*, 2014, 58, 314.). Hence, the higher calcination temperature may result in the formation of specific site to decrease in the H^+ transportation channels within the bulk.

In order to clarify these issues, we added above results to supplemental information and a relevant sentence as follows.

New [2-1]: Our result suggests that they also facilitate not only the H_2 dissociation step but also the H^+ diffusion steps in the hydrogen spillover process. **It should be noted that the higher calcination and reduction temperatures resulted in the sluggish of H^+ storage capacity because of the decrease of H^+ transportation channels within the bulk regions (Supplementary Fig. 12 and 13).** From the above results...

We further investigated the H^+ storage capacity of Ru/Al-MgO at different reduction temperatures to elucidate the effect of the Ru diameters. The diameters of Ru contained in the Ru/Al-MgO reduced at 600 and 800 °C was larger than that of Ru/Al-MgO reduced at 400 °C. As the reduction temperature of Ru/Al-MgO increased, the H^+ storage capacity decreased. Since the tendency of the reduction temperature in HD evolution experiments is consistent with that of calcination temperature, these results are considered to reflect the transformation of Al-MgO phase rather than the diameter of Ru nanoparticles. Hence, we would like to supplement these results with data showing that 400 °C is the optimum reduction temperature, rather than the influence of Ru particle

size.

Although we were unable to demonstrate the effect of Ru diameter accurately, we suppose that its effect is not crucial because we have previously reported that the activation energy for the dissociation of H₂ on Ru nanoparticle is much smaller compared with the subsequent spillover steps.

In order to suggest this, we added a following sentence.

New [2-2]: The mean diameters of Ru nanoparticles precipitated on MgO and Al-MgO were calculated to be 1.82 nm and 3.55 nm (**Supplementary Fig. 4**). We have previously reported that the dissociation of H₂ on Ru nanoparticles was a barrierless step compared with the subsequent migration steps¹⁷, and therefore excluded the effect on the diameter of Ru nanoparticles herein. Variation in mass of Ru/Al-MgO was evaluated...

3. It is odd that Ru-based catalysts were used to show the improved hydrogen spillover performance, while Pt and Ni-based catalysts were used to show the improved catalytic performance. How about the CO₂ hydrogenation performance of Ru-based catalysts? And the essential characterizations of Pt/Ni-based catalysts, such as HRTEM, XRD, H-D exchange, and H₂-TPD, should be provided.

Answer: Thank you very much for your useful comment.

We show the results in the CO₂ hydrogenation reaction using Ru/Ni based catalysts as below. As the results, the (Ru + Ni)/Al-MgO catalyst, a physical mixture of Ru/Al-MgO and Ni/Al-MgO, exhibited higher catalytic CO₂ conversion at all temperature ranges than its components, which is consistent to (Pt + Ni)/Al-MgO. Of these temperature ranges, (Ru + Ni)/Al-MgO exhibited 1.9-fold increase in activity enhancement compared with the average value of its components at 400 °C. This enhancement is moderate compared with that of (Pt + Ni)/Al-MgO which exhibited 4.6-fold increase. In this report,

we have exhibited the results using the Pt/Ni based catalysts because their activity enhancement by hydrogen spillover was more remarkable than Ru/Ni catalysts. This difference in activity enhancement is related to the kind of not platform materials but the deposited nanoparticles, therefore we will investigate this issue in detail in the next report.

In order to clarify this issue, we added above results to **Supplementary Fig. 17** and a sentence with yellow marker as below.

New [3-1]: We performed the same catalytic trials using various specimens with different designs: catalysts which employed Ru as the dissociation site of H₂ molecules (**Supplementary Fig. 17**), a catalyst which co-supported Pt and Ni (**Supplementary Fig. 18**), catalysts utilizing the Al-MgO calcined at different temperature as support materials (**Supplementary Fig. 19 and 20**), and catalysts with MgO as support material (**Supplementary Fig. 21**). Although the catalytic performances of these catalysts were superior to those of the individual catalysts respectively, their activity enhancements were moderate compared with that of (Pt + Ni)/Al-MgO shown in **Fig. 5b**. It has been reported that...

We further obtained structural characterizations for Pt/Ni catalysts.

From the XRD patterns, the Pt/Al-MgO and the Ni/Al-MgO exhibited diffractions attributable to only MgO and Ni. This result suggests that significant phase transformation did not occur during the deposition of Pt and Ni and the subsequent H₂ reduction. From TEM images the mean diameters of Pt and Ni on Al-MgO were calculated to be 9.26 and 49.3 nm, respectively.

Accordingly, we added above results to supplemental information and a sentence with yellow marker as below.

New [3-2]: In this catalyst, the Pt, Al-MgO and Ni promoted the dissociation of H₂ molecules, the transportation of atomic hydrogen from Pt to Ni sites and the utilization of atomic hydrogen for the CO₂ hydrogenation reaction, respectively. **No significant phase transformation occurred during the deposition of Pt and Ni and the subsequent H₂ reduction and the mean diameters of Pt and Ni nanoparticles on Al-MgO were 9.26 and 49.3 nm, respectively (Supplementary Fig. 16).** The major hydrogenated product...

4. The influence of Al content in Al-MgO sample on the hydrogen spillover effect should be provided.

Answer: Thank you very much for your constructive comment.

We investigated the H^+ storage capacity based on hydrogen spillover for the Ru/Al-MgO with various Mg/Al compositions. As the result, we obtained two insights. Firstly, the H^+ storage capacity for all Al-MgO was enhanced compared with the inherent MgO. Secondly, the Al-MgO with Mg/Al = 2 exhibited the largest H^+ storage capacity among all specimens. We estimate that the sample with an Mg/Al ratio of 2 showed the best H^+ storage capacity because it generated the largest amounts of specific sites, such as cation vacancy (V_{cat}) and tetrahedral-coordinated Al (Al_{Td}) within the original phase of MgO even though it exhibited some extent of phase transformation to spinel $MgAl_2O_4$, as shown **Fig. 1a**.

In this study, we want to focus on how the doped Al provide the H^+/e^- diffusion pathways with the MgO. In this context, the Al-MgO with an Mg/Al ratio = 2 exhibited significant phase separation which interferes with the accurate evaluation of the effect of Al in the original MgO phase. On the other hand, the Al-MgO with an Mg/Al ratio = 5 accommodated the maximum amount of Al without phase transformation. Therefore, we employed the Al-MgO with an Mg/Al ratio = 5 as a representative sample for the comparison with MgO.

Currently, we are developing the new synthesis method of Al-MgO which inhibits phase separation with large amount of Al dopant to improve the H^+ storage capacity. Therefore, we would like to discuss on the impact of Mg/Al composition in depth in the next work.

In order to clarify above issue, we provided above result with **Supplementary Fig. 5** and added following sentences.

New: Hence, the present material would be able to store an unprecedented amount of atomic hydrogen via H^+ diffusion even though it comprises Earth-abundant elements. Moreover, the promotional effect of Al doping on H^+ storage capacity was confirmed for the Al doped MgO with various Mg/Al compositions (**Supplementary Fig. 5**). In order to identify the specific H^+ transport channels within the bulk of MgO generated by the addition of a heteroatom Al, we investigated the H^+ diffusion property on the Al-MgO with the Mg/Al composition of 5 which accommodated maximum amount of Al without phase transformation. In these trials, the evolutions...

5. In Fig S8, it seems that at low temperatures (from 50 to 150 °C), hydrogen spillover on Al-MgO is not stronger than on MgO. Unlike that, hydrogen spillover on TiO₂ is also superior at room temperature. Does this limit its potential applications at low temperatures?

Answer: Thank you very much for your useful comment.

According to the *in situ* DRIFT measurements which we supplemented for **Answer1**, TiO₂ allows hydrogen spillover below 50 °C, whereas Al-MgO allows hydrogen spillover from 50 to 150 °C. Therefore, the hydrogen spillover on Al-MgO at room temperature is supposed to be limited compared with TiO₂, as you pointed. However, one the most important points we wanted to make in this paper is that the **H⁺ storage capacity** of Al-MgO derived from hydrogen spillover was superior to the typical reducible metal oxides even though there is possibility that reducible metal oxides exhibit superior hydrogen spillover performance in terms of other properties such as the migration distance and the occurrence temperature.

In order to clarify above issue, we modified some expressions as **New [1-1]** to **New [1-4]** in the **Answer1**.

6. It is better to provide a schematic diagram that includes both proton and electron migration paths. Does the motion of proton and electron in Al-MgO is coherent (because of charge balance)?

Answer: Thank you very much for your useful comment.

As you pointed out, we also think that the motion of H⁺ and e⁻ within the Al-MgO are coherent because the electron derived from hydrogen was not completely separated from the hydrogen atom when it biases to an adjacent Al, as shown **Fig. 3c**. However, we cannot experimentally prove whether e⁻ diffusion coupled with H⁺ on Al-MgO is identical to the diffusion in molecular hydrogen. Therefore, we would like to refrain from depicting e⁻ migration pathway in **Fig. 3d**.

7. Al-MgO was calcined at 400 °C under air for 1.25 h. The H-D experiment was carried out at 400 °C. And the CO₂ hydrogenation was carried out above 300 °C. The influence of calcination temperature of Al-MgO on hydrogen spillover and the catalytic performance should be added.

Answer: Thank you very much for your useful comment.

As we demonstrated in **Answer2**, the H⁺ storage capacity of Al-MgO decreases as the calcination temperature increases, owing to the decline of H⁺ transportation channels within the Al-MgO. These results were supplemented in **Supplementary Fig. 12**.

Additionally, we performed the catalytic CO₂ hydrogenation with Pt/Ni specimens whose support materials are Al-MgO calcined at 600 and 800 °C. As the result, these (Pt + Ni)/Al-MgO specimens also exhibited larger CO₂ conversion than mean values of corresponding Pt/Al-MgO and Ni/Al-MgO specimens, respectively. These tendencies are consistent to that of the specimens with Al-MgO calcined at 400 °C in **Fig. 5 (a) and (b)**. However, their activity enhancements of specimens with Al-MgO calcined at 600 and 800 °C were both 3.5, which were smaller than that of specimens using Al-

MgO calcined at 400 °C whose activity enhancement was 4.6.

Calculated at 600 °C

Calculated at 800 °C

In order to clarify above issue, we added a following sentence and supplemented above results in **Supplementary Fig. 19 and 20**.

New: We performed the same catalytic trials using various specimens with different designs; catalysts which employed Ru as the dissociation site of H₂ molecules (**Supplementary Fig. 17**), a catalyst which co-supported Pt and Ni (**Supplementary Fig. 18**), catalysts utilizing the Al-MgO calcined at different temperature as support materials (**Supplementary Fig. 19 and 20**), and catalysts with MgO as support material (**Supplementary Fig. 21**). Although the catalytic performances of these catalysts were superior to those of the individual catalysts respectively, their activity enhancements were moderate compared with that of (Pt + Ni)/Al-MgO shown in **Fig. 5b**. It has been reported that...

8. As to the WO₃ color change experiment, the H atoms may not migrate through the MgO/Al-MgO support, they could migrate from Ru particles to WO₃ directly. Is it a powerful characterization tool for electron diffusion?

Answer: Thank you very much for your useful comment.

We suppose that H⁺ and e⁻ migrate to WO₃ through the support materials rather than from Ru particles in this measurement.

In our results, there is almost no visible light absorption for WO₃ mixed with Ru/MgO, while WO₃ mixed with Ru/Al-MgO exhibited significant visible light absorption (**Fig. 4 a**). If H atoms directly migrate from Ru to WO₃, there should be no difference between the visible light absorptions of WO₃ mixed with Ru/MgO and Ru/Al-MgO. Several previous papers have studied the occurrence of hydrogen spillover with the similar strategy (*Nat. Catal.*, 2022, 5, 11, 1030-1037. *J. Am. Chem. Soc.*, 2019, 141, 21, 8482-8488.).

However, there is still possibility that the readers wonder if the H atoms directly migrate from Ru to WO₃. In order to avoid confusion, we added above references and some sentences with yellow highlighted as below.

New [8-1]: The hydrogen spillover ability of Al-MgO was further assessed in terms of e⁻ diffusion. WO₃ has been employed as a marker of hydrogen spillover in several reports^{37, 38}, because it shows change of its color from yellow to bronze by the acceptance of e⁻ due to the mixing of the valence transfer bands of W⁶⁺ and W⁵⁺.³⁹ Herein, the e⁻ diffusion associated with hydrogen spillover was studied from the visible light absorption of WO₃ mixed with Ru supported specimens under a H₂ reduction atmosphere with *in situ* UV-vis measurements (**Fig. 4a**). WO₃ mixed with the Ru/MgO showed no visible light absorption up to 400 °C...

New [8-2]: This result demonstrated that the H⁺ diffusion on the Al-MgO was accompanied by e⁻ migration, allowing hydrogen spillover to occur as in the case of reduced metal oxides even though Mg²⁺ and Al³⁺ are the least reducible. Note that H atoms do not migrate directly from Ru to WO₃ but through support materials, since there is no difference in the visible light absorptions of WO₃ mixed with Ru/MgO and Ru/Al-MgO. According to the electron spin resonance (ESR) spectra...

9. Is it accurate to calculate the average oxygen concentration within the Al-MgO from EDX line analysis results?

Answer: Thank you very much for your useful comment.

In order to confirm the atomic concentration of the Al-MgO, we further performed the inductively coupled plasma atomic emission spectroscopy (ICP-AES) measurement. As the result, the average atomic Mg:Al:O ratio of the Al-MgO was proven to be 38.2:10.0:51.8 which demonstrates the cation is still deficient in the Al-MgO. We suppose that the atomic ratio obtained from the EDX line analysis is also accurate, but it does not reflect the average concentration of the entire sample, but the local average concentration. Herein, the most important data should be provided is the information of the entire samples. Therefore, we would like to replace the result in **Fig. 1c** from EDX analysis to ICP-AES, and add the EDX analysis to supplemental information.

In order to clarify the above issue, we modified the **Fig. 1c** as below and provided the EDX line analysis in **Supplemental Fig. 3**.

Accordingly, the amount of V_{Cat} has been recalculated on the basis of ICP-OES and **Fig. 2** has been modified.

Moreover, we added the sentences regarding ICP-AES measurements with yellow highlighted as below.

New: The composition of Al_{Oh} and Al_{Td} were proven to be attributed at 80.8% and 19.2% within Al-MgO, respectively, with solid-state ^{27}Al magic angle spinning nuclear magnetic resonance (^{27}Al MAS NMR) spectroscopy (**Fig. 1b**). From the inductively coupled plasma atomic emission spectroscopy (ICP-AES) measurement, the Mg:Al:O ratio of the Al-MgO was proven to be 38.2:10.0:51.8, as shown in **Fig. 1c** even though the stoichiometric Mg:O ratio in pristine MgO is 1:1, which indicates the synthesized Al-MgO is in the cation deficient state. The energy dispersive X-ray spectroscopy (EDX) analysis provided the same tendency for atomic composition as ICP-AES and indicated that the Al was distributed in the nanometric region of Al-MgO (**Supplementary Fig. 3**). From the results of ICP-AES, the proportion of V_{Cat} at octahedral cation sites was calculated to be 9.2 % within Al-MgO (**Supplementary Note 1**). Hence, the incorporation of Al heteroatoms generates considerable number of two specific sites such as Al_{Td} and V_{Cat} within Al-MgO (**Fig. 2**).

10. In Fig. 5c, the quantitative analysis results of in situ XANES, such as linear fitting, should be provided.

Answer: Thank you very much for your useful comment.

We performed the linear fitting for the Ni K edge *in situ* XANES spectra of Ni/Al-MgO, (Pt + Ni)/MgO, and (Pt + Ni)/Al-MgO with that of Ni and NiO whose oxidation states are 0 and +2, respectively. As the results, the fraction of Ni⁰ was proven to be 0.35 for (Pt + Ni)/Al-MgO which is larger than Ni/Al-MgO by 0.22 under H₂ reduction atmosphere. This suggests that the reduction of Ni species is promoted by the presence of Pt. Additionally, this value was larger by 0.18 than that of (Pt + Ni)/MgO. This tendency was repeatedly obtained in the second H₂ dosage following to O₂ dosage. These results demonstrate that hydrogen spillover on Al-MgO significantly endows the reduction of Ni species and its reduction performance is superior to H⁺ diffusion on MgO.

In order to clarify above issue, we modified the **Fig. 5c** and added following sentences with yellow markers.

New: Hence, it is apparent that the Ni surfaces poisoned by oxygen atoms were readily reduced by the spilled hydrogen on Al-MgO. In order to confirm this, the redox behaviors of Ni species deposited on Ni/Al-MgO, (Pt + Ni)/MgO, and (Pt + Ni)/Al-MgO were examined from the linear fitting processing with Ni and NiO for the Ni K edge *in situ* X-ray absorption near edge structure (XANES) under alternating H₂ and O₂ atmospheres at 350 °C, respectively (**Fig. 5c**). As the results, the fraction of Ni⁰ calculated from Ni⁰/(Ni⁰ + Ni²⁺) was 0.35 for (Pt + Ni)/Al-MgO, which is larger by 0.22 than Ni/Al-MgO. This suggests that the reduction of Ni species is promoted by the presence of Pt under H₂ reduction atmosphere. Additionally, this value was larger by 0.18 than that of (Pt + Ni)/MgO. This

tendency was repeatedly obtained in the second H₂ dosage following to O₂ dosage. These results demonstrate that hydrogen spillover on Al-MgO significantly endows the reduction of Ni species and its reduction performance is superior to H⁺ diffusion on MgO. The results of H₂-TPR measurements...

Reviewer #2 (Remarks to the Author):

The present report illustrates that the proper introduction of cation vacancies in MgO by Al doping enables an efficient hydrogen spillover from metal sites to the non-reducible MgO support, which is otherwise thermodynamically unfavored. Although I read the manuscript with great interest and would like to recommend its publication in Nature Communications, I am afraid that the authors did not demonstrate the novelty of their findings adequately. My main concerns are addressed as follows.

Answer: We would like to thank reviewer 2 for the careful review and appreciate positive comments.

1. The authors highlighted in the abstract and introduction sections that hydrogen spillover is limited to occur over reducible metal oxides composed of rare elements. In the view of element abundance, graphitic structures and zeolites have already been reported for their application in hydrogen spillover (e.g., Chem. Rev. 2012, 112, 2714-2738; Nat. Commun. 2014, 5, 3370; Angew. Chem. Int. Ed. 2019, 58, 7668-7672). In addition, the ideas about introducing vacancy sites or semiconductors to promote hydrogen spillover have been comprehensively discussed by Roel Prins in his remarkable review (Chem. Rev. 2012, 112, 2714-2738). I am afraid the authors need to make it clear whether their findings are compatible with the existing knowledge or reflective of new strategies.

Answer: Thank you for your very constructive comment.

As you pointed out, the occurrence of hydrogen spillover on other platforms comprising the Earth-abundant elements and the promotion of hydrogen spillover by the presence of vacancy sites have been previously reported. However, we believe that our findings will also provide new insights into hydrogen spillover which were not discussed in above works, as follows.

First one is that the occurrence of hydrogen spillover was accomplished by a heteroatom doping strategy. As shown in **Fig 4**, the hydrogen spillover on the original MgO is inhibited, while the moderate Al doping allowed the occurrence of hydrogen spillover. This finding was not made in the conventional reports employing other platform materials comprising Earth-abundant elements and can further provide opportunities to produce new platforms for hydrogen spillover.

However, as you pointed, this manuscript does not clarify the difference from the previous reports using other platforms comprising Earth-abundant elements. Hence, we modified our manuscript to make it clear as follows.

In order to show that the non-reducible metal oxide is one of material groups comprising Earth-

abundant elements, we added a following sentence.

New [1-1]: Economically, metal oxides containing Earth-abundant elements represent more suitable platforms for hydrogen spillover. In this regard, employing non-reducible metal oxides, such as MgO and Al₂O₃, as a platform is one of candidates. Unfortunately, they inhibit the e⁻ diffusion due to their low reducibility...

In order to emphasize new insights into hydrogen spillover, we modified the title of this paper.

Old title: Hydrogen Spillover on a Non-reducible Metal Oxide Comprising Earth-abundant Elements

New title: Hydrogen Spillover on a Non-reducible Metal Oxide: Heteroatom Doping Strategy and H⁺/e⁻ Diffusion Pathways

Accordingly, we modified several sentences as follows.

New [1-2]: The present work demonstrates that a non-reducible MgO with heteroatom Al dopants (Al-MgO) allows hydrogen spillover in the same way as reducible metal oxides.

New [1-3]: The H⁺ and e⁻ diffusion pathways generated by the heteroatom Al doping was disentangled based on systematic characterizations and calculations.

New [1-4]: This work provides a new strategy for designing functional materials intended to hydrogen spillover for diverse applications in a future hydrogen-based society.

Second one is that cation vacancies provide the H⁺ transportation channels during the hydrogen spillover. As you pointed, it was previously reported in section 4.2 of the review by Roel Prins (*Chem. Rev.* 2012, 112, 2714-2738) that cation vacancies promote hydrogen spillover. In this review, it is mentioned that the cation vacancies react with H₂ which produce atomic H to initiate hydrogen spillover. In contrast the above discussion, we indicated that cation vacancies in the Al-MgO provided H⁺ transport channel to improve its H⁺ storage capacity. Such unique effect of cation vacancy on hydrogen spillover was not discussed in previous reports.

In order to emphasize new insights into effect of cation vacancies into hydrogen spillover, we added a following sentences.

New [1-5]: Second, the octahedral cation sites close to a V_{cat} have a 43.9% probability of being vacant in Al-MgO (**Supplementary Note 1**), which significantly increases A . It has been reported that V_{cat} can reacted with H_2 molecule to initiate hydrogen spillover.¹⁶ Our result suggests that they facilitate not only the H_2 dissociation step but also the H^+ diffusion steps in the hydrogen spillover process. It should be noted that...

2. The authors also emphasized that the Al-MgO sample they made exhibited superior hydrogen spillover performance. However, I am not sure how to define the superior performance. For instance, the authors did not compare Al-MgO with reducible metal oxides (e.g., CeO₂ and TiO₂) in hydrogen spillover.

Answer: Thank you very much for your constructive comment.

We have made two major revisions.

Firstly, we have changed several sentences.

In this paper, the most important hydrogen spillover performance we wanted to emphasize is that the **H⁺ storage capacity** of Al-MgO which was derived from hydrogen spillover was superior to typical reducible metal oxides. However, some expressions such as the sentence you pointed were inappropriate to clarify above issue because they can be taken to mean not H⁺ storage capacity but comprehensive hydrogen spillover ability of Al-MgO (e.g. rate and distance) is superior.

In order to avoid confusing, we modified corresponding sentences as follows.

Old [2-1]: The present work demonstrates that non-reducible MgO containing a moderate amount of Al (Al-MgO), comprising Earth-abundant elements, exhibits superior hydrogen spillover performance. This material has a H storage capacity 3.1 times greater than those of various standard metal oxides based on hydrogen transport channels within its bulk region.

New [2-1]: The present work demonstrates that non-reducible MgO containing a moderate amount of Al (Al-MgO), comprising Earth-abundant elements, **allows hydrogen spillover in the same way as reducible metal oxides. Furthermore, a H⁺ storage capacity of this materials owing to hydrogen spillover** was 3.1 times greater than those of various standard metal oxides based on hydrogen transport channels within its bulk region.

Old [2-2]: The design of materials based on such common elements that also provide significant hydrogen spillover ability will enable the development of innovative and sustainable technologies for a next-generation hydrogen society.

New [2-2]: The design of materials based on such common elements that also **allow** hydrogen spillover **ability** will enable the development of innovative and sustainable technologies for a next-generation hydrogen society.

Old [2-3]: The present work demonstrates that Al-doped MgO (Al-MgO), which contains only Earth-abundant elements, exhibits excellent hydrogen spillover performance.

New [2-3]: The present work demonstrates that a non-reducible Al-doped MgO (Al-MgO), which contains only Earth-abundant elements, allows hydrogen spillover in the same way as reducible metal oxides.

Old [2-4]: The non-reducible Al-MgO exhibited superior hydrogen spillover ability despite being composed of only Earth-abundant elements.

New [2-4]: The Al-MgO allowed hydrogen spillover that is the coupled H^+ and e^- diffusion and exhibited superior H^+ storage capacity to typical reducible metal oxides even though it comprises only Earth-abundant elements.

Secondly, we have added a supplemental figure and relevant a sentence to provide data concerning hydrogen spillover rate.

For other properties, we suppose that the occurrence temperature of the Al-MgO is superior to CeO_2 and WO_3 . The hydrogen spillover occurred around 50, 150, and 250 °C on the Ru-supported TiO_2 , CeO_2 , and WO_3 by *in situ* DRIFT measurements speculating O–D bonds derived from D^+ diffusion. In this regard, the Ru/Al-MgO exhibited hydrogen spillover from 50 to 150 °C, therefore Al-MgO allows the hydrogen spillover at lower temperature than CeO_2 and WO_3 .

In order to clarify above issue, we added *in situ* DRIFT measurements for Ru-supported TiO_2 , CeO_2 and WO_3 together with the Al-MgO and a relevant sentence as follows.

New [2-5]: The area ratios of these peaks to the first peak were calculated to be 5.6 and 5.9, respectively, even though it was only 1.3 in the case of the second peak generated from Ru/MgO, showing that superior H⁺ storage capacity of Al-MgO was originated from the two H⁺ transport channels within the bulk region. Note that H⁺ diffusion on the Al-MgO occurs at lower temperature than CeO₂ and WO₃ and therefore the diffusion rate is supposed to be superior to these two reducible metal oxides (Supplementary Fig. 9). To identify specific channels within the Al-MgO...

3. It is noticeable that the authors assessed the proton and electron diffusion properties of the Al-MgO sample separately, while the hydrogen spillover requires atomic H species migrate along solid surfaces via coherent proton-electron movements. In particular, Roel Prins has clearly elucidated that the H⁺ diffusion itself does not correlate with hydrogen spillover (Chem. Rev. 2012, 112, 2714-2738), because H⁺ diffusion can occur readily with H⁺ exchange between surface hydroxyl groups. Therefore, I don't think the content about the proton diffusion is very helpful.

Answer: Thank you very much for your constructive comment.

As you pointed, hydrogen spillover is the coherent proton-electron movement and therefore an elucidation of only H⁺ diffusion is not helpful for the demonstration of hydrogen spillover. Hence, we explored the hydrogen spillover property of Al-MgO from the viewpoint of e⁻ diffusion property in addition to H⁺ diffusion property.

In order to further emphasize this, we added following sentences with yellow markers.

New [3-1]: Atomic hydrogen diffuses over the non-reducible Al-MgO produced active H⁺-e⁻ pairs, as also occurs on reducible metal oxides, to enhance the catalytic performance of Ni during CO₂ hydrogenation. The H⁺ and e⁻ diffusion pathways generated by the heteroatom Al doping was disentangled based on systematic characterizations and calculations. This work provides insights...

New [3-2]: The present work demonstrates that Al-doped MgO (Al-MgO), which contains only Earth-abundant elements, allows hydrogen spillover in the case of reducible metal oxides comprising rare elements. The hydrogen spillover property of Al-MgO was evaluated in terms of H⁺ diffusion and e⁻ diffusion by utilizing a variety of characterizations and theoretical calculations. In this material...

New [3-3]: Hence, the incorporation of Al heteroatoms generates considerable number of two specific sites such as Al_{Td} and V_{Cat} within Al-MgO (Fig. 2). Based on this structure, we investigated the hydrogen spillover, the coupled H⁺ and e⁻ diffusion, on the Al-MgO based on the diffusion of H⁺ and e⁻.

4. Related with the previous issue, the authors discussed the H⁺ transport channel in the bulk of Al-MgO. However, hydrogen spillover is well known as a surface phenomenon. It is suggested to focus more on the surface properties, such as surface compositions and the concentration of surface vacancy sites.

Answer: Thank you very much for your constructive comment.

As you pointed, the hydrogen spillover phenomenon has been widely well known as a surface phenomenon. However, several papers have recently reported the hydrogen spillover at not surface region, such as subsurface and bulk of platform materials (*J. Phys. Chem. C*, 2009, 113, 26, 11399. *Catal. Sci. Technol.*, 2018, 8, 22, 5750. *Nat. Commun.*, 2018, 9, 1, 3778. *Top Catal.*, 2022, 65, 1, 270.) and it can also endow the enhancement of catalytic activity (*Nat. Commun.*, 2019, 10, 1, 4166.). Therefore, we suppose that hydrogen spillover into the bulk region is a crucial viewpoint for the evaluation of hydrogen spillover.

However, our current manuscript may confuse the readers because hydrogen spillover is widely known as a surface phenomenon. In order to emphasize the significance of hydrogen spillover into the bulk region, we added following sentences before we start the discuss on the hydrogen spillover into the bulk.

New: The proportional H⁺ storage on the Al-MgO was calculated to be 0.29 wt% and so was 3.1 times larger than that obtained using the MgO in spite of its less than half BET surface area (S_{BET}) (**Fig. 3a**). Recently, several reports have indicated that hydrogen spillover occurred in the bulk region of metal oxide platforms²⁸⁻³⁰ and it also can endow the catalysis.³¹ Considering that Al-MgO exhibited larger H⁺ storage capacity even though its S_{BET} was smaller than MgO...

5. In my view, one of the most convincible pieces of evidence about the occurrence of hydrogen spillover on oxide-supported metal catalysts is the reversible formation of surface hydroxy species when the catalyst is exposed to H₂ at elevated temperatures (*Nat. Catal.* 2013, 6, 710-719). Have the authors observed such phenomena in their DRIFT studies on Ru/Al-MgO and Ru/MgO?

Answer: Thank you very much for your constructive comment.

In order to obtain insights into the reversible formation of hydroxyl species, we performed *in situ* DRIFT measurements under the H₂-D₂ switching atmosphere for Ru/MgO and Ru/Al-MgO. Both specimens exhibited the formation of $\delta_{\text{O-D}}$ during the D₂ dosage, while it was attenuated under the H₂ dosage. These DRIFT results are the clear evidence of the hydrogen spillover.

In order to clarify above issue, we added above result to the **Supplementary Fig. 8**.

6. About the part of catalytic CO₂ hydrogenation, the authors are strongly suggested to compare the catalytic performance at a constant temperature instead of using the transient experiments shown in Figure 5a, because the steady-state performance is more importance to catalysis. Moreover, what result will be obtained if Pt and Ni are co-loaded on the same Al-MgO support for catalytic CO₂ hydrogenation?

Answer: Thank you very much for your crucial pointing.

In Fig. 5a, we compared the catalytic performance not at transient temperature but at constant temperatures. In detail, we hold the catalysts at target temperature for 10 minutes before checking the catalytic performance. In order to avoid the misunderstanding, we provided additional information in the experimental section.

New: The reaction products generated at 300, 350, 400, 450 and 500 °C were analyzed online using a gas chromatograph (Shimadzu GC-14B) and employing an activated carbon column connected to a thermal conductivity detector followed by a flame ionization detector equipped with a methanizer. Each catalyst was held at the target temperature for 10 minutes before checking the catalytic performance.

Moreover, we performed an additional catalytic trial using an Al-MgO on which Pt and Ni are co-loaded specimen (labeled as PtNi/Al-MgO). As the result, the PtNi/Al-MgO also exhibited superior performance during this reaction to the mean value of the Pt/Al-MgO and the Ni/Al-MgO. However, the activity enhancement of the co-loaded PtNi/Al-MgO was 2.8 and surprisingly inferior to physically mixed (Pt + Ni)/Al-MgO.

In order to clarify above issue, we supplemented above results to **Supplementary Fig. 18** and added following sentence.

New: We performed the same catalytic trials using various specimens with different designs: catalysts which employed Ru as the dissociation site of H₂ molecules (**Supplementary Fig. 17**), a catalyst which co-supported Pt and Ni (**Supplementary Fig. 18**), catalysts utilizing the Al-MgO calcined at different temperature as support materials (**Supplementary Fig. 19 and 20**), and catalysts with MgO as support material (**Supplementary Fig. 21**). Although the catalytic performances of these catalysts were superior to those of the individual catalysts respectively, their activity enhancements were moderate compared with that of (Pt + Ni)/Al-MgO shown in **Fig. 5b**. It has been reported that...

We estimate that two possibilities why the activity enhancement was sluggish even though Pt and Ni was supported on the same Al-MgO. First possibility is the formation of Pt and Ni alloy nanoparticles. As shown in **Fig. 5d**, we propose that the role of Pt is the dissociation of H₂ to drive hydrogen spillover, whereas that of Ni is the activation of CO₂ on the (Pt + Ni)/Al-MgO. In this regard, the alloying could rather lose each inherent properties of Pt and Ni due to geometric and electronic effects. Second possibility is that the hydrogen spillover effect is enlarged as the migration distance increases. In our unpublished work, metal ions loaded on a platform oxide different from Pt were reduced at lower temperatures than metal ions loaded on the same platform oxide as Pt. In this regard, there is a possibility that the redox of the Ni species on (Pt + Ni)/Al-MgO significantly proceeded compared with the Ni species on PtNi/Al-MgO owing to the enlarged hydrogen spillover effect.

Reviewer #3 (Remarks to the Author):

Kazuki Shun and co-workers reported that non-reducible MgO containing a moderate amount of Al (Al-MgO), comprising Earth-abundant elements, exhibits superior hydrogen spillover performance. The diffusion of H⁺ and e⁻ within Al-MgO were studied by systematic characterizations and calculations. However, the work lacks key experimental results to support their conclusion.

Answer: We would like to thank reviewer 3 for the careful review.

1. page3_line64-66 the authors mentioned that the Mg/Al ratio of 5 is used as the optimal ratio. However, you did not explain the relationship between hydrogen spillover and the ratio of Mg/Al. Please provide more experiment results to support the Mg/Al ratio of 5 as the optimal ratio or give a trend of hydrogen spillover performance with different Mg/Al ratios.

Answer: Thank you very much for your constructive comment.

We investigated the H⁺ storage capacity based on hydrogen spillover for the Ru/Al-MgO with various Mg/Al compositions. As the result, we obtained two insights. Firstly, the H⁺ storage capacity for all Al-MgO was enhanced compared with the inherent MgO. Secondly, the Al-MgO with Mg/Al = 2 exhibited the largest H⁺ storage capacity among all specimens. We estimate that the sample with an Mg/Al ratio of 2 showed the best H⁺ storage capacity because it generated the largest amounts of specific sites, such as cation vacancy (V_{Cat}) and tetrahedral-coordinated Al (Al_{Td}) within the original phase of MgO even though it exhibited some extent of phase transformation to spinel MgAl₂O₄, as shown **Fig. 1a**.

In this study, we want to focus on how the doped Al provide the H⁺/e⁻ diffusion pathways with the MgO. In this context, the Al-MgO with an Mg/Al ratio = 2 exhibited significant phase separation

which interferes with the accurate evaluation of the effect of Al in the original MgO phase. On the other hand, the Al-MgO with an Mg/Al ratio = 5 accommodated the maximum amount of Al without phase transformation. Therefore, we employed the Al-MgO with an Mg/Al ratio = 5 as a representative sample for the comparison with MgO.

Currently, we are developing the new synthesis method of Al-MgO which inhibits phase separation with large amount of Al dopant to improve the H⁺ storage capacity. Therefore, we would like to discuss on the impact of Mg/Al composition in depth in the next work.

In order to clarify above issue, we provided above result with **Supplementary Fig. 5** and added following sentences.

New: Hence, the present material would be able to store an unprecedented amount of atomic hydrogen via H⁺ diffusion even though it comprises Earth-abundant elements. **Moreover, the promotional effect of Al doping on H⁺ storage capacity was confirmed for the Al doped MgO with various Mg/Al compositions (Supplementary Fig. 5).** In order to identify the specific H⁺ transport channels within the bulk of MgO generated by the addition of a heteroatom Al, we investigated the H⁺ diffusion property on the Al-MgO with the Mg/Al composition of 5 which accommodated maximum amount of Al without phase transformation. In these trials, the evolutions...

2. page 3_line 76, authors used EDX to analyze the content of Al.

Please use inductively coupled plasma atomic emission spectroscopy (ICP-AES) to give Al and Mg content. Without accurate content, it is impossible to prove your conclusion including the proportion of the cation vacancies.

Answer: Thank you very much for your useful comment.

In order to confirm the atomic concentration of the Al-MgO, we further performed the inductively coupled plasma atomic emission spectroscopy (ICP-AES) measurement. As the result, the average atomic Mg:Al:O ratio of the Al-MgO was proven to be 38.2:10.0:51.8 which demonstrates the cation is still deficient in the Al-MgO. We suppose that the atomic ratio obtained from the EDX line analysis is also accurate but it exhibits not the average concentration of the entire sample, but the local average concentration. Herein, the most important data should be provided is information of the entire samples. Therefore, we would like to replace the result in Fig. 1c from EDX analysis to ICP-AES, and add the EDX analysis to supplemental information.

In order to clarify above issue, we modified Fig. 1c as below and provided the EDX line analysis in Supplemental Fig. 3.

Accordingly, the amount of V_{Cat} has been recalculated on the basis of ICP-OES and Fig. 2 has been modified.

Moreover, we added the sentences regarding ICP-AES measurements with yellow markers as below.

New: The composition of Al_{Oh} and Al_{Td} were proven to be attributed at 80.8% and 19.2% within Al-MgO, respectively, with solid-state ^{27}Al magic angle spinning nuclear magnetic resonance (^{27}Al MAS NMR) spectroscopy (**Fig. 1b**). From the inductively coupled plasma atomic emission spectroscopy (ICP-AES) measurement, the Mg:Al:O ratio of the Al-MgO was proven to be 38.2:10.0:51.8, as shown in **Fig. 1c** even though the stoichiometric Mg:O ratio in pristine MgO is 1:1, which indicates the as-synthesized Al-MgO is in the cation deficient state. The energy dispersive X-ray spectroscopy (EDX) analysis provided the same tendency for atomic composition as ICP-AES and indicated that the Al was distributed in the nanometric region of Al-MgO (**Supplementary Fig. 3**). From the results of ICP-AES, the proportion of V_{Cat} at octahedral cation sites was calculated to be 9.2 % within Al-MgO (**Supplementary Note 1**). Hence, the incorporation of Al heteroatoms generates considerable number of two specific sites such as Al_{Td} and V_{Cat} within Al-MgO (**Fig. 2**).

3 Fig. S3. I cannot accept analyzing TEM data like this. Please improve the quality of your TEM data, especially the analysis process.

Please get FFT pattern transformed from your image and then an inverse Fourier image transformed from FFT pattern by using different pairs of lattice points can be used to explain your results.

In addition, please provide the TEM data of MgO.

Answer: Thank you very much for your constructive comment.

In order to explain our results, we obtained Williamson-hall plots and FFT patterns for MgO and Al-MgO.

From the slope of Williamson-hall plots obtained from XRD patterns of MgO and Al-MgO, we compared the amount of internal strain (*Solid State Sci.*, 2011, 13, 251.). As the result, the overall amount of strain within Al-MgO was 3.1 times larger than MgO. This information supports the formation of specific sites such as Al_{Td} and V_{Cat} within Al-MgO which provide lattice strain. It should be noted that MgO we synthesized also has lattice strain owing to its poor crystallinity derived from low calcination temperature, but its amount is smaller than Al-MgO.

In order to obtain additional information, we obtained an inverse FFT image for a region of interest (ROI) for Al-MgO by using systematic diffraction spots of 001 and 010. As the result, the inverse FFT of Al-MgO exhibited the bending of lattice. Therefore, we estimate that the presence of Al_{Td} and V_{Cat} appeared as the dislocation within Al-MgO. In some regions, the similar bending was also observed for MgO as shown below, however, the frequency of distortion was lower than Al-MgO which is consistent to above Williamson-hall plots.

In order to clarify above issue, we added the Williamson-hall plot and the ROI and inverse FFT of Al-MgO to **Supplementary Fig. 2**, respectively and a following sentence with yellow marker.

New: It has been reported that cation vacancies (V_{Cat}) produce in periclase MgO to maintain the charge balance by the substitution of trivalent Al^{3+} in periclase. **Considering the larger amount of internal strain within Al-MgO than MgO as revealed by the Williamson-hall plots, it is likely that the specific sites such as Al_{Td} and V_{Cat} was introduced within Al-MgO (Supplementary Fig. 2).** We quantified the concentration of...

In addition to above information, we have evaluated the presence of these specific sites within Al-MgO from various measurements such as the XRD (**Fig. 1a**), the solid state MAS-NMR (**Fig. 1b**), the ICP-AES (**Fig. 1c**), and the EDX analysis (**Supplementary Fig.3**). Furthermore, the formation of V_{Cat} and Al_{Td} in the Al doped MgO has been previously reported (e.g., *J. Catal.*, 1998, 178, 499-510. *Phys. Earth Planet. Inter.*, 2009, 172, 34-42. *J. Am. Chem. Soc.*, 1994, 116, 1707-1717. *ChemSusChem*, 2019, 12, 2810-2818.) Considering the consistency with above characterizations and previous reports, we concluded that the V_{Cat} and Al_{Td} certainly exist within Al-MgO.

In order to clarify above issue, we added the following references to corresponding sentences.

[25] Hiremath, V. et al. Mg-Ion Inversion in MgO@MgO–Al₂O₃ Oxides: The Origin of Basic Sites. *ChemSusChem* **12**, 2810-2818 (2019).

[27] Van Orman, J. A., Li, C. & Crispin, K. L. Aluminum diffusion and Al-vacancy association in periclase. *Phys. Earth Planet. Inter.* **172**, 34-42 (2009).

4. Fig. S3 and S4 provided almost the same information. Please explain the reason.

For EDX mapping, please provide the image of Mg+O+Al. From your results, Al seems more concentrated in some places.

Answer: Thank you for your important pointing.

We provided two EDX mappings in **Supplementary Fig. 3** and **4** because this information exhibited not entire structure but local structure of Al-MgO.

However, as you pointed, the information provided in these figures are almost the same and there is possibility that it makes confuse. Therefore, we removed **Supplementary Fig. 4**.

Besides, we added the image of Mg+Al+O in **Supplementary Fig. 3**. The Al is distributed in the Al-MgO particle without significant phase separation in the nanometric region. However, this result cannot completely exclude the possibility that there are some places where the Al is concentrated in the angstrom microscopic areas. Therefore, we emphasize that the Al was distributed **in the nanometric region**.

In order to clarify above issue, we added the following sentences with yellow markers.

New: From the inductively coupled plasma atomic emission spectroscopy (ICP-AES) measurement, the Mg:Al:O ratio of the Al-MgO was proven to be 38.2:10.0:51.8 as shown in **Fig. 1c** even though the stoichiometric Mg:O ratio in pristine MgO is 1:1, which indicates the as-synthesized Al-MgO is

in the cation deficient state. The energy dispersive X-ray spectroscopy (EDX) analysis provided the same tendency for atomic composition as ICP-AES and indicated that the Al was distributed in the nanometric region of Al-MgO (Supplementary Fig. 3). From the results of ICP-AES...

5 Fig. S6. Please explain why the authors used the TEM image for Ru/ MgO but the HAADF-STEM image for Ru/Al-MgO.

Answer: Thank you for raising an important point.

As the focus depth of STEM image is shallower than that of TEM images, the length of objects on the different height obtained from HAADF-STEM images is tend to be overestimated. We used TEM images to obtain size distribution for Ru/MgO because the deposited Ru particles on the support was clear. On the other hand, we used HAADF-STEM images because it was difficult to identify the deposited Ru particles from TEM images as shown below. The difficulty is owing to the diffraction contrast derived from the abundant internal strain within Al-MgO that is associated with the presence of specific sites such as Al_{Td} and V_{Cat} . Therefore, we used the TEM image for Ru/ MgO but the HAADF-STEM image for Ru/Al-MgO.

A TEM image of Ru/Al-MgO specimen.

6 Fig. S6. Please explain how to identify which particles are Ru particles in the HAADF-STEM image. For the CO₂ hydrogenation part, I am confused about the experimental design of the authors.

Answer: We identified the Ru particles from HAADF-STEM images considering the Z-contrast.

In the HAADF-STEM images, the intensity is considered to be proportional to Z^2 value due to Rutherford scattering where z is atomic number. In this term, the Z^2 value of Ru is 1936, which is 13.4, 11.5, and 30.3 times larger than that of Mg, Al, and O, respectively. Moreover, we selected the field of view where the edges of bright particles are sharper. In this region, the contrast of support material's area is uniform which demonstrates that the total support thickness of Al-MgO is almost constant. In this way, we distinguished the Ru particles from Al-MgO support. Also, the locations of the brighter particles in the HAADF-STEM image were roughly consistent to the region where the concentration of Ru L was relatively high.

Additionally, we carried out additional experiments and added some critical discussions for the effect of hydrogen spillover on the catalysis CO₂ hydrogenation by using various catalysts as well as Pt and Ni co-deposited catalyst. Please see the Answer to the comment 3 from reviewer 1, the comment 7 from reviewer 1, and the comment 6 from reviewer 2 for details.

7 page 7 lines 204-205 Authors mentioned that Hence, it is apparent that the Ni surfaces poisoned by oxygen atoms were readily reduced by the spilled hydrogen on Al-MgO.

Two samples (Pt+Ni)/Al-MgO (i) and Ni/Al-MgO (ii) are under CO₂ hydrogenation condition to check X-ray absorption near edge structure of Ni species. The change of oxidized Ni (ii) and oxidized Ni (i) should be provided.

8 page 7 lines 214 and 215

Authors should provide results of the control sample (Pt+Ni)/MgO under alternating H₂ and O₂ atmospheres to support your conclusion.

Answer to 7 & 8: Thank you very much for your useful comment.

We performed the linear fitting for the Ni K edge *in situ* XANES spectra of Ni/Al-MgO, (Pt + Ni)/MgO, and (Pt + Ni)/Al-MgO with that of Ni and NiO whose oxidation states are 0 and +2. As the results, the fraction of Ni⁰ was proven to be 0.35 for (Pt + Ni)/Al-MgO which is larger than Ni/Al-MgO by 0.22 under H₂ reduction atmosphere which exhibits that the reduction of Ni species is promoted by the presence of Pt. Additionally, this value was larger by 0.18 than that of (Pt + Ni)/MgO. This tendency was repeatedly obtained in the second H₂ dosage following to O₂ dosage. These results demonstrate that hydrogen spillover on Al-MgO significantly endows the reduction of Ni species and its reduction performance is superior to H⁺ diffusion on MgO.

In order to clarify above issue, we modified the **Fig. 5 (c)** and added following sentences with yellow markers.

New: Hence, it is apparent that the Ni surfaces poisoned by oxygen atoms were readily reduced by the spilled hydrogen on Al-MgO. In order to confirm this, the redox behaviors of Ni species

deposited on Ni/Al-MgO, (Pt + Ni)/MgO, and (Pt + Ni)/Al-MgO were examined from the linear fitting processing with Ni and NiO for the Ni K edge *in situ* X-ray absorption near edge structure (XANES) under alternating H₂ and O₂ atmospheres at 350 °C, respectively (Fig. 5c). As the results, the fraction of Ni⁰ calculated from Ni⁰/(Ni⁰ + Ni²⁺) was 0.35 for (Pt + Ni)/Al-MgO, which is larger by 0.22 than Ni/Al-MgO which exhibits that the reduction of Ni species is promoted by the presence of Pt under H₂ reduction atmosphere. Additionally, this value was larger by 0.18 than that of (Pt + Ni)/MgO. This tendency was repeatedly obtained in the second H₂ dosage following to O₂ dosage. These results demonstrate that hydrogen spillover on Al-MgO significantly endows the reduction of Ni species and its reduction performance is superior to H⁺ diffusion on MgO. The results of H₂-TPR measurements...

REVIEWERS' COMMENTS

Reviewer #1 (Remarks to the Author):

The authors have extensively revised their manuscript and have explained the changes they made in great detail. The authors have addressed my comments. I now recommend publication.

Reviewer #2 (Remarks to the Author):

I would like to appreciate the authors greatly for their detailed and thoughtful response to my comments. The authors have made a significant improvement in this revised version. However, I am still a little concerned about the H₂-D₂ switching experiments mentioned in their response to my fifth comment.

In my view, the formation of the O-D vibrotational signal during the D₂ dosage may merely reflect the exchange of the D atom formed on the metal surface and the proton in the surface hydroxyl groups of the oxide support, considering the ubiquitous hydroxyl groups on the MgO and Al-MgO surfaces. As clearly demonstrated by Prof. Prins (Chem. Rev. 2012, 112, 2714–2738), H–D Exchange is not proof of spillover. What the authors should confirm is whether the total concentration of surface hydroxyl groups can reversibly increase when the catalyst is exposed in the atmosphere of H₂.

Reviewer #3 (Remarks to the Author):

The authors have addressed most of my concerns.

We would like to thank all reviewer for the careful review and the valuable comments, which allowed us to improve the paper. Below we list the changes we have made in light of the reviewer's comments.

Reviewer #1 (Remarks to the Author):

The authors have extensively revised their manuscript and have explained the changes they made in great detail. The authors have addressed my comments. I now recommend publication.

Answer: We would like to thank reviewer 1 for valuable comments in review process.

Reviewer #2 (Remarks to the Author):

I would like to appreciate the authors greatly for their detailed and thoughtful response to my comments. The authors have made a significant improvement in this reversed version. However, I am still a little concerned about the H₂-D₂ switching experiments mentioned in their response to my fifth comment.

In my view, the formation of the O-D vibrotational signal during the D₂ dosage may merely reflect the exchange of the D atom formed on the metal surface and the proton in the surface hydroxyl groups of the oxide support, considering the ubiquitous hydroxyl groups on the MgO and Al-MgO surfaces. As clearly demonstrated by Prof. Prins (Chem. Rev. 2012, 112, 2714–2738), H–D Exchange is not proof of spillover. What the authors should confirmed is whether the total concentration of surface hydroxyl groups can reversibly increase when the catalyst is exposed in the atmosphere of H₂.

Answer: We appreciate your detailed explanation and an important pointing.

We performed *in situ* DRIFT measurement under D₂ and Ar switching atmosphere for Ru/Al-MgO to monitor reversible increase of the total concentration of surface O–D groups. As the result, the intensity of O–D peak increased during D₂ dosage and attenuated during inert Ar dosage. Notably, this result was reproducible from the second trials onwards. The reason for using D₂ instead of H₂ is that it was difficult to identify an increase or decrease in the signal of the O–H groups ($\delta_{\text{O-H}}$) because we could not completely remove H₂O, which competes the $\delta_{\text{O-H}}$, from the infrared light path in our DRIFT system.

In order to clarify above issue, the results of *in situ* DRIFT spectra between H₂ and D₂ provided in **Fig. S8** was replaced above result.

Also, as the referee pointed out, only H–D Exchange is not proof of hydrogen spillover, the simultaneous diffusion of H⁺ and e⁻. Therefore, we herein examined e⁻ diffusion property as well as H⁺ diffusion.

We would like to thank reviewer 2 for valuable comments in review processes.

Reviewer #3 (Remarks to the Author):

The authors have addressed most of my concerns.

Answer: We would like to thank reviewer 3 for valuable comments in review process.